# Gain-of-function mutations in the UNC-2/CaV2α channel lead to excitation-dominant synaptic transmission in *Caenorhabditis elegans*

Yung-Chi Huang[1], Jennifer K Pirri[1], Diego Rayes[1†], Shangbang Gao[2‡], Ben Mulcahy[2], Jeff Grant[1], Yasunori Saheki[3§], Michael M Francis[1], Mei Zhen[2,4,5], Mark J Alkema[1]*

[1]Department of Neurobiology, University of Massachusetts Medical School, Worcester, United States; [2]Lunenfeld-Tanenbaum Research Institute, Mount Sinai Hospital, Toronto, Canada; [3]Lulu and Anthony Wang Laboratory of Neural Circuits and Behavior, The Rockefeller University, New York, United States; [4]Department of Molecular Genetics, University of Toronto, Toronto, Canada; [5]Department of Physiology, University of Toronto, Toronto, Canada

*For correspondence:
mark.alkema@umassmed.edu

Present address: †Instituto de Investigaciones Bioquímicas de Bahía Blanca (CONICET), Departamento de Biología, Bioquímica y Farmacia, Universidad Nacional del Sur, Bahía Blanca, Argentina; ‡College of Life Science and Technology, Huazhong University of Science and Technology, Wuhan, China; §Lee Kong Chian School of Medicine, Nanyang Technological University, Nanyang, Singapore

Competing interests: The authors declare that no competing interests exist.

**Abstract** Mutations in pre-synaptic voltage-gated calcium channels can lead to familial hemiplegic migraine type 1 (FHM1). While mammalian studies indicate that the migraine brain is hyperexcitable due to enhanced excitation or reduced inhibition, the molecular and cellular mechanisms underlying this excitatory/inhibitory (E/I) imbalance are poorly understood. We identified a gain-of-function (gf) mutation in the *Caenorhabditis elegans* CaV2 channel α1 subunit, UNC-2, which leads to increased calcium currents. *unc-2(zf35gf)* mutants exhibit hyperactivity and seizure-like motor behaviors. Expression of the *unc-2* gene with FHM1 substitutions R192Q and S218L leads to hyperactivity similar to that of *unc-2(zf35gf)* mutants. *unc-2(zf35gf)* mutants display increased cholinergic and decreased GABAergic transmission. Moreover, increased cholinergic transmission in *unc-2(zf35gf)* mutants leads to an increase of cholinergic synapses and a TAX-6/calcineurin-dependent reduction of GABA synapses. Our studies reveal mechanisms through which CaV2 gain-of-function mutations disrupt excitation-inhibition balance in the nervous system.
DOI: https://doi.org/10.7554/eLife.45905.001

## Introduction

Maintenance of proper brain function requires the balance of excitatory and inhibitory synaptic transmission. There is an increasing amount of evidence that the disruption of E/I balance in neural circuits is associated with neurological disorders, including autism, epilepsy and migraine (*Nelson and Valakh, 2015*; *Vecchia and Pietrobon, 2012*). Several studies have proposed that impaired inhibitory function may drive a shift in E/I balance toward excitation, and underlie the phenotypic changes observed in these disorders (*Selten et al., 2018*; *Mainero and Louapre, 2014*). While animal model studies provide support for this hypothesis, our understanding of the molecular and cellular mechanisms that lead to E/I imbalance remains limited.

Mutations in the *CACNA1A* gene, which encodes the pore-forming α subunit of the CaV2.1 (P/Q-type) voltage-gated calcium channel (VGCC), are associated with a broad spectrum of autosomal dominant neurological disorders. CaV2 VGCCs are the predominant channels in presynaptic nerve terminals, where they mediate the $Ca^{2+}$ influx that triggers the fusion of synaptic vesicles with the presynaptic membrane (*Catterall, 2000*; *Bidaud et al., 2006*). *CACNA1A* mutations can cause

episodic ataxia type 2 (EA2), epileptic seizures and familial hemiplegic migraine type 1 (FHM1) (*Pietrobon, 2010*). Episodic ataxia type 2 (EA2), whose clinical features include the lack of voluntary coordination of muscle movements and epileptic seizures, is associated with a range of missense, nonsense-, and splice site mutations throughout the *CACNA1A* gene. Familial hemiplegic migraine type 1 (FHM1), a severe variant of migraine that can co-occur with tonic-clonic seizures, has been found to be associated with missense mutations near the voltage sensors of the α1 subunit (*Adams and Snutch, 2007*). Electrophysiological analyses suggest that EA2 mutations lead to diminished channel functions, whereas both gain- and loss-of-channel function phenotypes have been reported for FHM1-associated mutations (*Cao et al., 2004*; *Tottene et al., 2002*). Although these disorders have been conventionally distinguished, they exhibit considerable overlap in clinical presentations, leaving a precise correlation between genotype and phenotype unresolved.

Animal model studies can provide mechanistic insights into the pathology of *CACNA1A* mutations. Mice carrying FHM1 missense mutations R192Q or S218L in the *cacna1a* gene display gain-of-function CaV2 phenotypes with increased $Ca^{2+}$ current density at lower voltages (*Tottene et al., 2009*; *Zhou et al., 2017*; *van den Maagdenberg et al., 2010*). In FHM1 knock-in mice, glutamatergic neurotransmission in cortical pyramidal cells is enhanced, while GABAergic neurotransmission is unaltered. These findings suggest that FHM1 mutations cause a dysregulation of cortical E/I balance (*Vecchia and Pietrobon, 2012*).

The genome of the nematode *Caenorhabditis elegans* encodes a single CaV2α subunit gene: *unc-2* (*Schafer and Kenyon, 1995*). UNC-2/CaV2α is exclusively expressed in the nervous system (*Mathews et al., 2003*) and localizes to presynaptic zones, at synaptic vesicle release sites (*Saheki and Bargmann, 2009*), as well as at the plasma membrane of neural somas (*Gao et al., 2018*). Behaviorally, *unc-2* loss-of-function (lf) mutants are sluggish and uncoordinated (*Mathews et al., 2003*). Furthermore, *unc-2(lf)* mutants have a reduced frequency of spontaneous excitatory postsynaptic currents (EPSCs) (*Richmond et al., 2001*), and a reduced intrinsic neuronal calcium oscillations of *C. elegans* motor neurons (*Gao et al., 2018*).

In this study, we characterize a novel *unc-2/CaV2α* gain-of-function (gf) mutant, which, in sharp contrast to the loss-of-function mutant, exhibits hyperactive- as well as seizure-like motor behaviors. We show that the expression of an *unc-2* gene carrying FHM1 mutations results in a similar hyperactive behavioral phenotype, while the intragenic suppressor alleles of *unc-2(gf)* resemble EA2 mutations and are lethargic. We reveal that the *unc-2(gf)* mutation shifts the E/I balance toward excitatory transmission, and that increased excitatory signaling leads to the destabilization of GABAergic synapses in a TAX-6/calcineurin-dependent manner.

## Results

### *zf35* mutants are hyperactive

*C. elegans* locomotion is biased toward sustained forward runs, interrupted by periodic brief reversals. From a forward genetic screen for animals with locomotion defects, we isolated a mutant, *zf35*, which failed to execute sustained forward or backward runs and continually switched the direction of locomotion in a jerky manner (reversal frequency: *zf35*: 43.1 ± 2.0/3 min, n = 59; wild type: 6.8 ± 0.4/3 min, n = 59) (*Figure 1*; *Video 1*). This clonic seizure-like phenotype of *zf35* mutants was accompanied by an increased locomotion rate during bouts of forward or backward locomotion. On average, *zf35* mutants moved approximately 1.5 fold faster than wild-type animals (*Figure 1A and B*). Animals heterozygous for the *zf35* mutation also displayed increased velocity and reversal frequency (*Figure 1A–C*), albeit to a lesser extent when compared to homozygous mutants. This indicates that the *zf35* mutation is semi-dominant. *zf35* mutant animals were slightly smaller than wild-type animals (0.82 ± 0.03 mm, n = 75 vs 1.00 ± 0.04 mm, n = 88) (*Figure 1D*) and had a reduced brood size (wild type: 207 ± 11, n = 5, *zf35*: 150 ± 16, n = 5). Furthermore, *zf35* adults retained a reduced number of eggs in the uterus (*zf35*: 3.6 ± 0.2, n = 86; wild type: 14.1 ± 0.6, n = 80) (*Figure 1E*). *zf35* mutants laid eggs that are at an earlier developmental stage than wild-type animals, indicating that the time between fertilization and egg laying was reduced (*Figure 1F*). Therefore, *zf35* mutants are hyperactive in both locomotion and egg-laying behaviors.

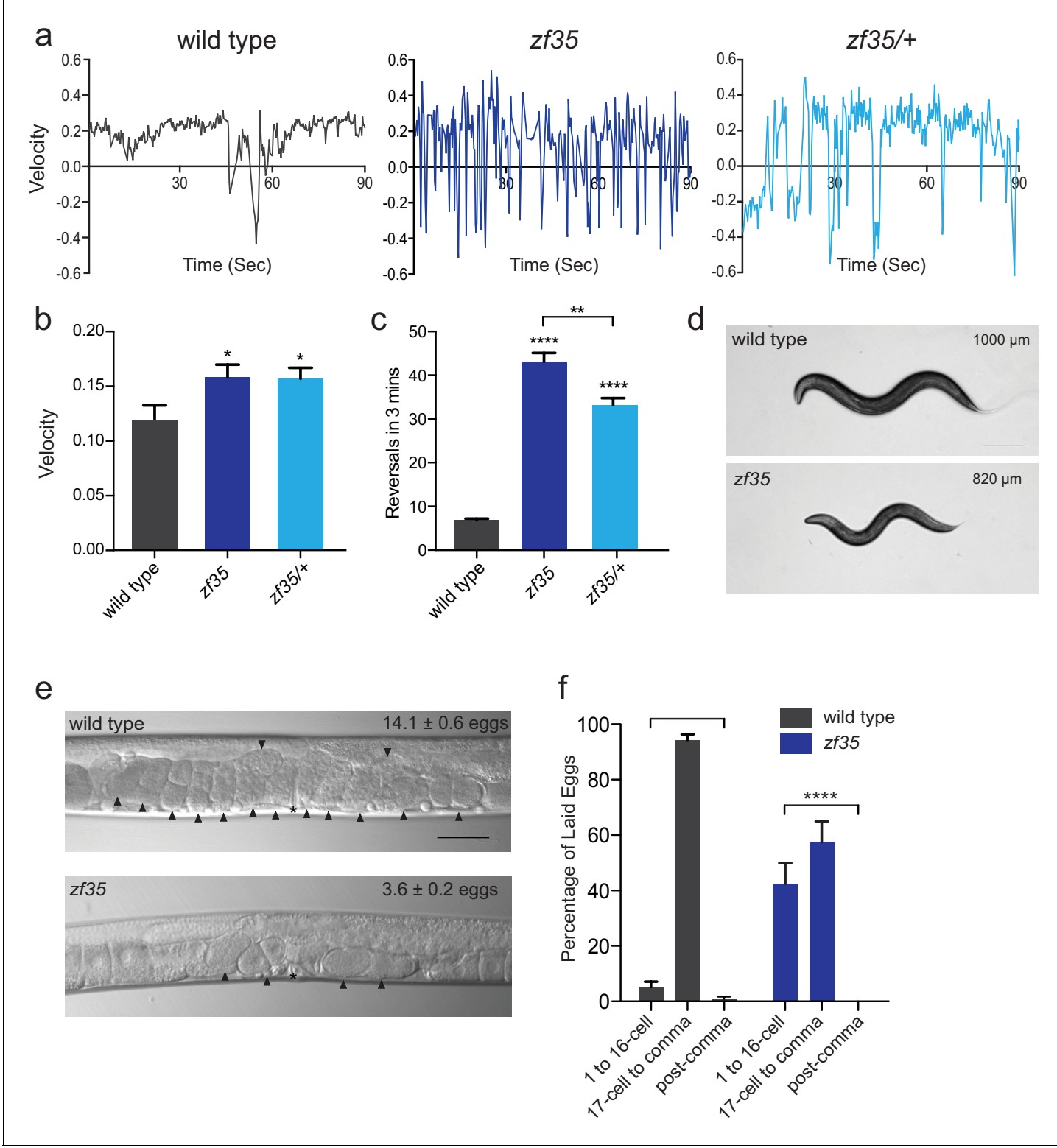

**Figure 1.** *zf35* animals are hyperactive in both locomotion and egg-laying behaviors. (**a**) Representative traces from single worm tracking showing instantaneous velocity of indicated genotypes on OP50 thin lawn plates (see Materials and methods). Positive and negative values indicate forward and backward locomotion, respectively. Transition from positive to negative values indicates reversal events. (**b**) Shown is the average velocity for the wild-type (0.118 ± 0.01 worm lengths/s, n = 9), *zf35* (0.156 ± 0.01 worm lengths/sec, n = 10), *zf35* /+ (0.155 ± 0.01 worm lengths/s, n = 10) animals (**c**) Quantification of the reversal frequency in 3 min on regular OP50 plates: average reversal numbers made by wild type (6.8 ± 0.4 reversals, n = 59), *zf35*

*Figure 1 continued on next page*

*Figure 1 continued*

(43.1 ± 2.0 reversals, n = 59) and *zf35/+* (33.2 ± 1.6 reversals, n = 23). Error bars represent SEM for at least three trials. Statistical difference from wild type *p<0.05, ****p<0.0001, one-way ANOVA with Dunnett's multiple comparisons test. Statistical difference between *zf35* and *zf35/+* **p<0.01, unpaired t-test. (d) Representative images of wild type and *zf35* animals. Average of midline lengths of the wild type: 1.00 ± 0.04 mm, n = 88 and *zf35*: 0.82 ± 0.03 mm, n = 75. Scale bar is 200 µm. (e) Representative Nomarski images of unlaid eggs in adult wild-type and *zf35* animals. Arrowheads indicate eggs; asterisk denotes the position of the vulva. The average numbers of eggs in the uterus: wild type (14.1 ± 0.6 eggs, n = 80), *zf35* (3.6 ± 0.2 egg, n = 86) animals. Scale bar, 50 µm. (f) Embryonic stages of freshly laid eggs of the wild type and *zf35* mutants. 43% of the laid eggs from *zf35* animals are at 1–16 cell stage, while only 5% from the wild type laid eggs are at 1–16 cell stage. Five independent trials with 75 animals for each genotype. Statistical difference from wild type ****p<0.0001, Chi-squared test.

DOI: https://doi.org/10.7554/eLife.45905.002

The following source data is available for figure 1:

**Source data 1.** Source data for *Figure 1*.
DOI: https://doi.org/10.7554/eLife.45905.003

## *zf35* mutant's hyperactivity is caused by a missense mutation in the *unc-2/CaV2α* gene

We mapped the *zf35* mutation to the left end of chromosome X between genetic markers *lon-2* and *dpy-3*. This region contains a gene, *unc-2*, which encodes the α1 subunit of the *C. elegans* CaV2 voltage-gated calcium channel. Sequencing analysis of the *zf35* allele revealed a single-base transition (GGAto AGA) in the 17th exon of *unc-2* (*Figure 2A*). UNC-2/CaV2α consists of four homologous domains (I-IV) each containing six hydrophobic membrane-spanning segments (S1–S6, *Figure 2—figure supplement 1*). The *zf35* mutation results in a glycine to arginine substitution (G1132R) in the highly conserved intracellular linker between III-S6 and IV-S1 (*Figure 2B and C*). To determine if UNC-2(G1132R) in *zf35* mutant animals is sufficient to confer the hyperactive phenotype, we generated an *unc-2(zf35)* cDNA clone, which encodes the UNC-2/CaV2α(G1132R) protein. Pan-neuronal expression of the *unc-2(zf35)* transgene, in both wild-type and *unc-2* loss-of-function mutant (lf) backgrounds, induced hyperactive behavior similar to that of the *zf35* mutant. Transgenic overexpression of the wild-type *unc-2* cDNA rescued the uncoordinated and lethargic phenotype of *unc-2 (lf)* mutants, but did not induce hyperactive behavior (*Figure 2E*). *unc-2(zf35)* mutants did not display obvious defects in neural morphology (data not shown). To determine if the *zf35* mutation affected UNC-2 localization, we generated transgenic animals carrying C-terminus GFP-tagged *unc-2 (zf35)* cDNA. UNC-2(GF/G1132R)::GFP was observed in the cell soma, and in puncta along the neuronal processes (*Figure 2D*). The fluorescence expression pattern of UNC-2(GF/G1132R)::GFP animals displayed no obvious difference with that of a UNC-2(WT)::GFP transgene (*Figure 2D*) (*Saheki and Bargmann, 2009*). This indicates that UNC-2(G1132R) is properly processed and trafficked to presynaptic sites.

## Intragenic mutations suppress the *unc-2(zf35)* hyperactivity phenotype

*unc-2* loss-of-function (lf) mutants are sluggish and exhibit reduced motor activities (*Mathews et al., 2003*). *unc-2(lf)* mutants are also slightly longer than wild type animals, most likely due to reduced muscle contraction. The contrasting phenotypes between *unc-2(zf35)* and *unc-2(lf)* mutants suggested that the *zf35* G1132R mutation is a rare gain-of-function

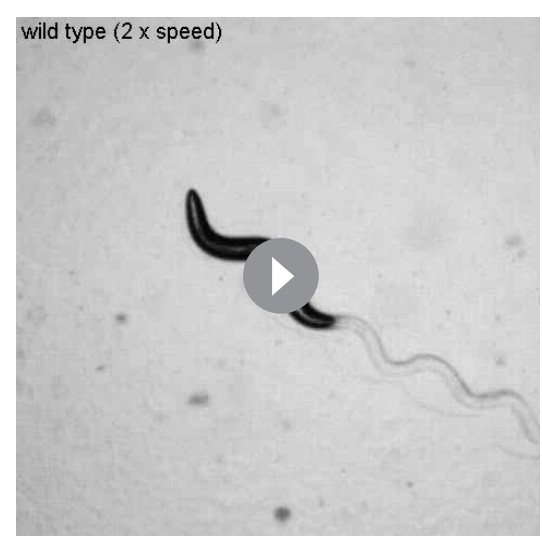

**Video 1.** *unc-2(zf35)* mutants have an increased reversal frequency. Videos of locomotor behavior of the wild-type and *unc-2(zf35)* animals on NGM agar plates with seeded OP50.
DOI: https://doi.org/10.7554/eLife.45905.004

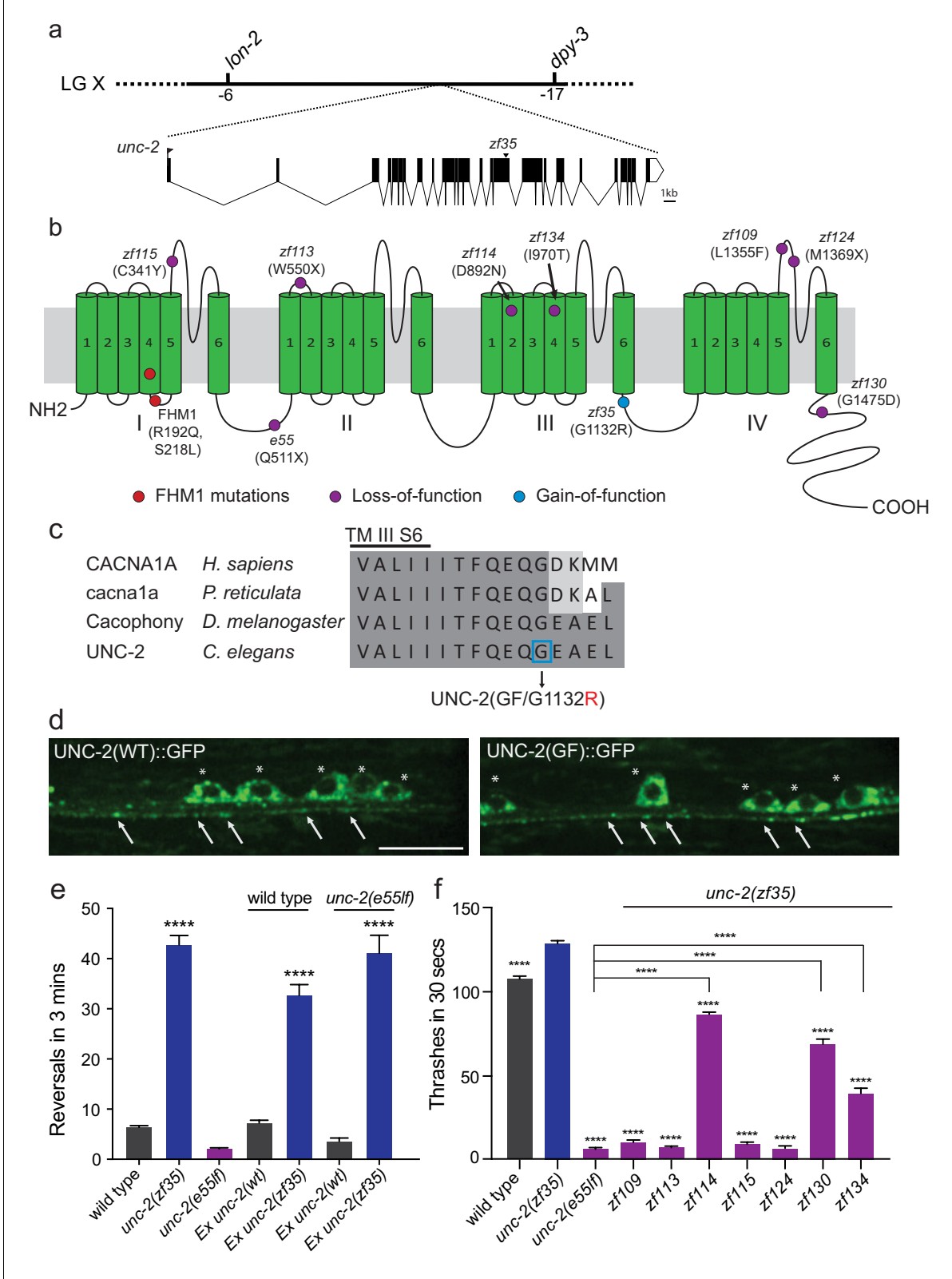

**Figure 2.** *zf35* is a novel allele of the CaV2α subunit gene *unc-2*. (a) The genetic map and gene structure of *unc-2*. Coding sequences are represented as black boxes. The *zf35* allele is a single nucleotide transition (<u>G</u>GA to <u>A</u>GA) resulting in a glycine to arginine (G to R) amino acid substitution at position 1132. (b) Diagram of the secondary structure of UNC-2/CaV2α. UNC-2/CaV2α consists of four domains (I–IV) each containing six alpha-helix transmembrane (TM) segments (S1 – S6). The UNC-2 (G1132R) mutation localizes in the intracellular loop between TM domain III and IV, indicated by

*Figure 2 continued on next page*

*Figure 2 continued*

the blue circle. Purple circles indicate positions of intragenic *unc-2(zf35)* suppressors, red circles indicate the location of human FHM1 mutations. (**c**) The G1132R mutation occurs in a highly conserved region of the CaV2α subunit. Amino acid alignment of C-terminus region of the transmembrane III alpha-helix segment 6 (III S6) and the beginning of the third intracellular loop of CaV2α subunits from human (*Homo sapiens*, CACNA1A), rainbow fish (*Poecilia reticulata*, cacna1a), fly (*Drosophila melanogaster*, Cacophony) and nematode (*C. elegans*. UNC-2). Identities are shaded in dark gray, similarities in light gray. Location of the G1132R mutation is indicated. (**d**) Representative images of GFP tagged UNC-2(WT) and UNC-2(GF/G1132) in the ventral nerve cord. Asterisks point the cell bodies of the motor neurons and arrows indicate the presynaptic sites. Both constructs are expressed under pan-neuronal promoter *tag-168*. Scale bar, 10 μm. (**e**) Quantification of the reversal frequency: wild type (6.6 ± 0.4, n = 70), *unc-2(zf35)* (43.3 ± 1.9, n = 65), *unc-2(e55lf)* (2.4 ± 0.2, n = 59), wild-type animals expressing *unc-2(wt)* transgene (7.5 ± 0.6, n = 10) and *unc-2(zf35)* transgene (33 ± 2.1, n = 22), and *unc-2(e55lf)* rescued with *unc-2(wt)* transgene (3.8 ± 0.7, n = 12) and *unc-2(zf35)* transgene (41.3 ± 3.6, n = 21). Error bars represent SEM for at least three trials with indicated totaling animals number. Statistical difference from wild type ****p<0.0001, one-way ANOVA with Dunnett's multiple comparisons test. (**f**) Intragenic *unc-2(lf)* mutations suppress *unc-2(zf35)* hyperactive locomotion. Shown are numbers of thrashes in 30 s in M9 for the wild type (107.0 ± 14.0, n = 60), *unc-2(zf35)* (128.1 ± 13.5, n = 60), *unc-2(lf)* (4.8 ± 2.1, n = 57), *unc-2(zf35 zf109)* (6.9 ± 4.3, n = 53); *unc-2(zf35 zf113)* (5.6 ± 3.7 thrashes, n = 57); *unc-2(zf35 zf114)* (80.2 ± 9.9, n = 60); *unc-2(zf35 zf115)* (6.9 ± 3.8, n = 56); *unc-2(zf35 zf124)* (5.3 ± 3.1, n = 57); *unc-2(zf35 zf130)* (67.1 ± 22.5, n = 58); *unc-2(zf35 zf134)* (31.2 ± 17.9, n = 50). Error bars represent SEM. Statistical difference from *unc-2(zf35)* mutants unless otherwise indicated, ****p<0.0001, one-way ANOVA with Tukey's multiple comparisons test.

DOI: https://doi.org/10.7554/eLife.45905.005

The following source data and figure supplement are available for figure 2:

**Source data 1.** Source data for *Figure 2*.
DOI: https://doi.org/10.7554/eLife.45905.007
**Figure supplement 1.** Amino acid alignment of human CACNA1A and *C. elegans* UNC-2 proteins.
DOI: https://doi.org/10.7554/eLife.45905.006

mutation. If so, secondary, loss-of-function mutations in the *unc-2* locus should function as intragenic suppressors of the hyperactivity phenotype of *unc-2(zf35)*. From a screen of mutagenized *unc-2(zf35)* mutants, we identified seven intragenic suppressor alleles that harbor missense or non-sense secondary mutations in the *unc-2* gene (*Figure 2B*).

Four suppressors, *zf109, zf113, zf115* and *zf124* reverted the *zf35* hyperactivity phenotype to sluggish locomotion, similar to the canonical loss-of-function *unc-2(e55)* allele (*Figure 2F*). The *zf113* (W550stop) and *zf124*(M1369stop) alleles result in premature stop codons and therefore likely represent null alleles of *unc-2*. The *zf115*(C341Y) and *zf109*(L1355F) missense mutations result in substitutions of conserved amino acids in the S5-S6 loop of domain I and IV, respectively. Two suppressors, *zf134* and *zf130*, caused moderate locomotion defects. The *zf134*(I970T) mutation affects an amino acid in the conserved voltage sensor, and the *zf130*(G1475D) mutation affects the C-terminal region, between a conserved EF-hand and the IQ-like motif. One suppressor, *zf114*(D892N), which changes an amino acid in the domain III S2, restored locomotion behavior of *unc-2(zf35)* to approximately wild-type levels. These intragenic suppressors represent an allelic series of hypomorphic *unc-2* mutations. Their ability to revert the hyperactive phenotype of *zf35* mutants to that of the wild-type or *unc-2(lf)* mutants strongly suggest that the *zf35* mutation is a gain-of-function allele of *unc-2*. Therefore, from here on the *unc-2(zf35)* allele will be referred to as *unc-2(zf35gf)*.

## A G to R substitution in CaV2α intracellular III-IV linker leads to increased CaV2 channel activity

To investigate the functional consequences of the UNC-2/CaV2α G1132R gain-of-function mutation, we introduced the corresponding change (G1518R) into the human P/Q type CaV2.1 channel α1 subunit, CACNA1A (*Figure 2C*). CaV2.1α expression constructs were transfected into a HEK 293 cell line that stably expresses the auxiliary β1c and α2δ subunits (*Piedras-Renteria et al., 2001*). Whole-cell patch clamp experiments (*Figure 3A*) showed that the CACNA1A(G1518R) CaV2.1α channel exhibited a −10 mV shift in activation potential when compared to the wild-type CaV2.1α channel (*Figure 3B and C*). The maximal current density was 1.7-fold larger for G1518R channels (80.6 ± 5.7 pA/pF, n = 11) compared to wild type (47.5 ± 4.3 pA/pF, n = 13) (*Figure 3B*).

The slope of the activation curve was not significantly affected in the CACNA1A(G1518R) channel ($K_a$WT = 3.8 ± 0.2 mV; $K_a$ G1518R = 4.1 ± 0.1 mV, *Figure 3C*). Both wild-type and G1518R CaV2.1 channels decayed with similar mono-exponential time courses ($T_{inac}$ CACNA1A(wt)=177 ± 45 ms and $T_{inac}$ CACNA1A(G1518R)=196 ± 32 ms at a 0 mV pulse). This suggests that the transition from the open to the inactive states was not affected by the G1518R mutation. To determine if inactivation

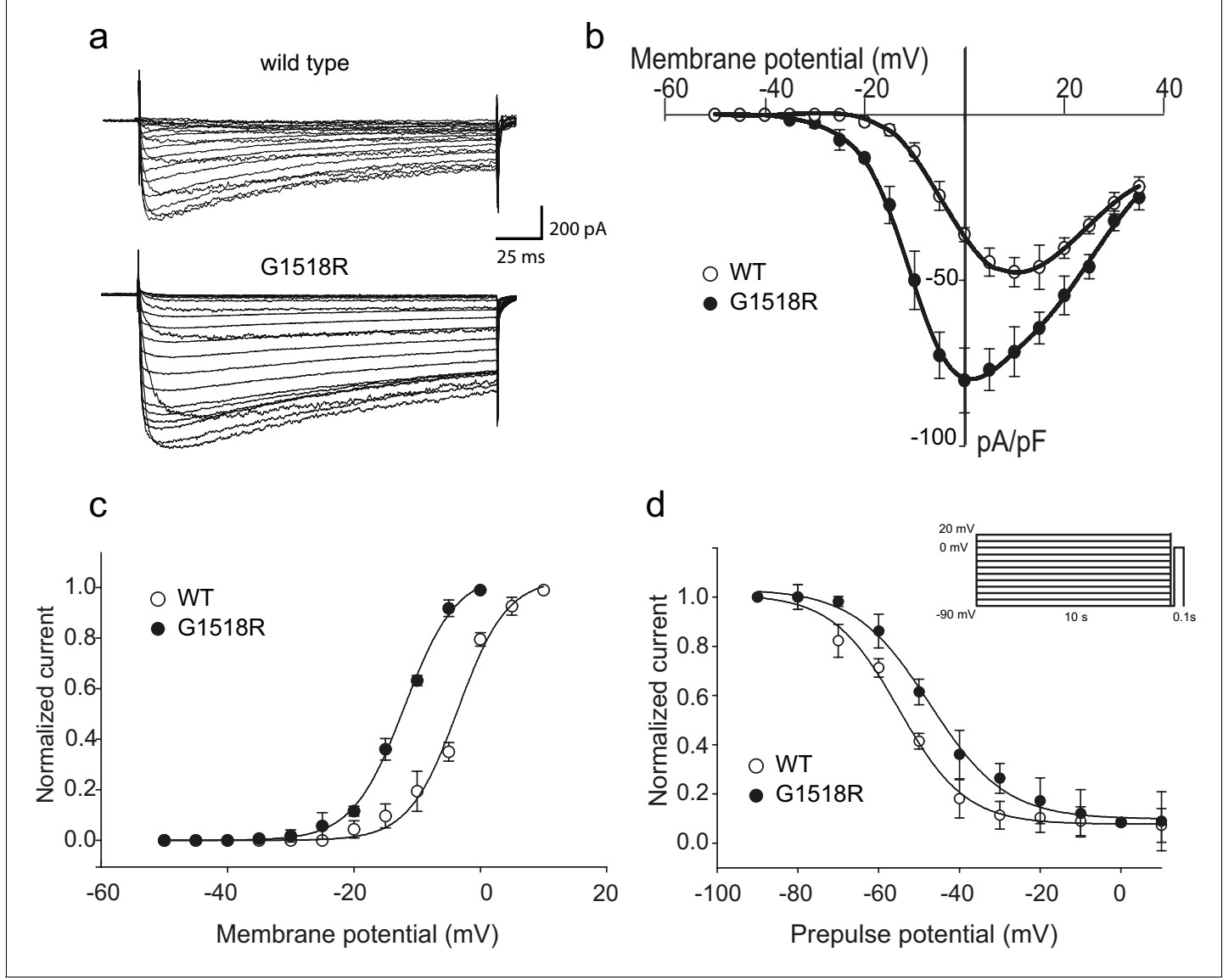

**Figure 3.** The UNC-2(G1132R) corresponding mutation in human CACNA1A, CaV2.1α, subunit results in increased channel activity. (a) Representative macro-currents of wild type and G1518R CaV2.1 channels. Currents were generated by stepping membrane potential to voltages between −55 and 40 mV in 5 mV increments for 200 ms from a holding potential of −120 mV. (b) Voltage dependence of whole-cell current density for wild type and G1518R CaV2.1 channels. Current density values were obtained by dividing current amplitudes and cell capacitance. (Wild type, n = 13; G1518R, n = 11). (c) Voltage dependence of Ba$^{2+}$ current activation. The activation curve of G1518R exhibits a significant shift of the V$_{0.5}$ value towards more negative membrane potentials. (d) Steady-State inactivation curves. The G1518R mutation causes a slight positive shift in the midpoint voltage in the steady-state inactivation curves (V$_{0.5inact}$= -55.0 ± 1.0 and −47.3 ± 1.0 for wild type and G1518R, respectively). Currents were normalized to the maximal value obtained at the test pulse and plotted as a function of the prepulse potential. Data were fitted with the Boltzmann equation: ($I_{max}$=(1+exp[(V-V0.5)/kin]) - 1). All recordings were carried out in Ba$^{2+}$ solution to exclude the effects from calcium-dependent inactivation.

DOI: https://doi.org/10.7554/eLife.45905.008

The following source data is available for figure 3:

**Source data 1.** Source data for *Figure 3*.
DOI: https://doi.org/10.7554/eLife.45905.009

following closed states was altered, we compared steady-state inactivation properties of wild-type and G1518R channels (*Figure 3D*). The membrane potential at which half of the current was inactivated in the G1518R channels exhibited a 7.7 mV shift to more positive potentials compared to wild type (V$_{0.5inact}$= -55.0 ± 1.0 mV and −47.3 ± 1.0 mV for the wild-type and G1518R channels,

respectively). This displacement indicates that the proportion of activatable channels is increased for CACNA1A(G1518R) channels at a given membrane potential. Thus, the G1518R mutation leads to channels that are activated at lower membrane potentials, and inactivated at higher membrane potentials. Together, these properties lead to increased current density by CACNA1A(G1518R). The conservation in the linker between TM III and TM IV between *C. elegans* and mammals suggests that UNC-2(G1132R) exhibits similar gain-of-function effects in activation and inactivation kinetics of CaV2α channel. However, since these experiments were performed with the human CACNA1A channel in HEK cells, we cannot exclude the possibility that the corresponding UNC-2/CaV2α G1132R mutation may have different effects on channel function in *C. elegans*.

## FHM1-analogous mutations in UNC-2/CaV2α lead to behavioral hyperactivity

Several missense mutations in the human *CACANA1A* gene result in familial hemiplegic migraine type 1 (FHM1) (*Pietrobon, 2010*). Electrophysiological analyses of the effects on CaV2.1 channel kinetics of FHM1 mutations in heterologous expression systems vary considerably and can even be contradictory. For instance, while some reports find that the R192Q mutation decreases CaV2.1 calcium transients (*Cao et al., 2004*; *Tottene et al., 2002*), others find that the same mutation results in an increased calcium influx at lower membrane potentials (*Hans et al., 1999*; *van den Maagdenberg et al., 2004*). In knock-in mouse models, the R192Q and S218L, FHM1 mutations increased $Ca^{2+}$ current density indicating a gain-of-function effect (*van den Maagdenberg et al., 2004*; *Tottene et al., 2009*; *van den Maagdenberg et al., 2010*). To determine the effects of FHM1 mutations in *C. elegans*, we introduced analogous R192Q and S218L mutations into *unc-2* (*Figure 4A*, *Figure 2—figure supplement 1*). Pan-neuronal expression of the *unc-2(R192Q)* or *unc-2(S218L)* transgene in *C. elegans* resulted in phenotypes similar to *unc-2(zf35gf)* mutants. Specifically, both *unc-2(R192Q)* and *unc-2(S218L)* animals exhibited increased reversal frequencies (25.5/min ±0.9, n = 34 and 16.5/min ±0.9, n = 33, respectively) when compared to wild-type animals (4.2/min ±0.5, n = 29) (*Figure 4B*). They also displayed hyperactive egg-laying behavior (*Figure 4C*). *unc-2(FHM1)* transgenic animals laid eggs that are at an earlier developmental stage and retained fewer eggs in the uterus (*unc-2(R192Q)*: 5.7 ± 0.4, n = 37; *unc-2(S218L)*: 8.4 ± 0.6, n = 32, respectively), when compared to wild-type animals (16.5 ± 0.8, n = 23). These experiments provide strong genetic evidence that, similar to *unc-2(zf35)*, the FHM1 mutations are gain-of-function mutations that lead to increased CaV2 activity.

## *unc-2/CaV2α* gain-of-function mutations increase sensitivity to aldicarb

Our electrophysiological recordings suggested the UNC-2(GF/G1132R) channel may increase $Ca^{2+}$ influx, resulting in elevated neurotransmitter release. To assess if *unc-2(zf35gf)* mutants have altered synaptic transmission, we analyzed their sensitivity to the acetylcholinesterase inhibitor, aldicarb. *C. elegans* body wall muscles receive input from excitatory cholinergic motor neurons (*White et al., 1986*; *Richmond and Jorgensen, 1999*). Aldicarb treatment causes the accumulation of acetylcholine (ACh), inducing muscle hypercontraction and acute paralysis (*Miller et al., 1996*). Approximately 50% of wild-type animals exposed to 1 mM aldicarb became paralyzed within 1 hr, consistent with previous findings (*Miller et al., 1996*; *Mathews et al., 2003*), while less than 15% *unc-2(lf)* mutants were paralyzed within 1 hr (*Figure 4D*). In sharp contrast, almost 100% of *unc-2(zf35gf)* mutants became paralyzed within 30 min (*Figure 4D*). Heterozygous *unc-2(zf35gf)/+* mutants also paralyzed more rapidly than the wild-type, confirming that the *unc-2(zf35gf)* mutation is semi-dominant.

Pan-neuronal expression of *unc-2(R192Q)* or *unc-2(S218L)* also induced hypersensitivity to aldicarb (*Figure 4E*). This hypersensitivity is not due to overexpression of the *unc-2* transgene because expression of a wild-type *unc-2* transgene, which restored the locomotion defects in *unc-2(lf)* mutants, led to wild-type sensitivity to aldicarb (*Figure 4E*). Therefore, animals with gain-of-function mutations in *unc-2/CaV2α* are hypersensitive to aldicarb, which may reflect increased ACh release at the neuromuscular junction (NMJ).

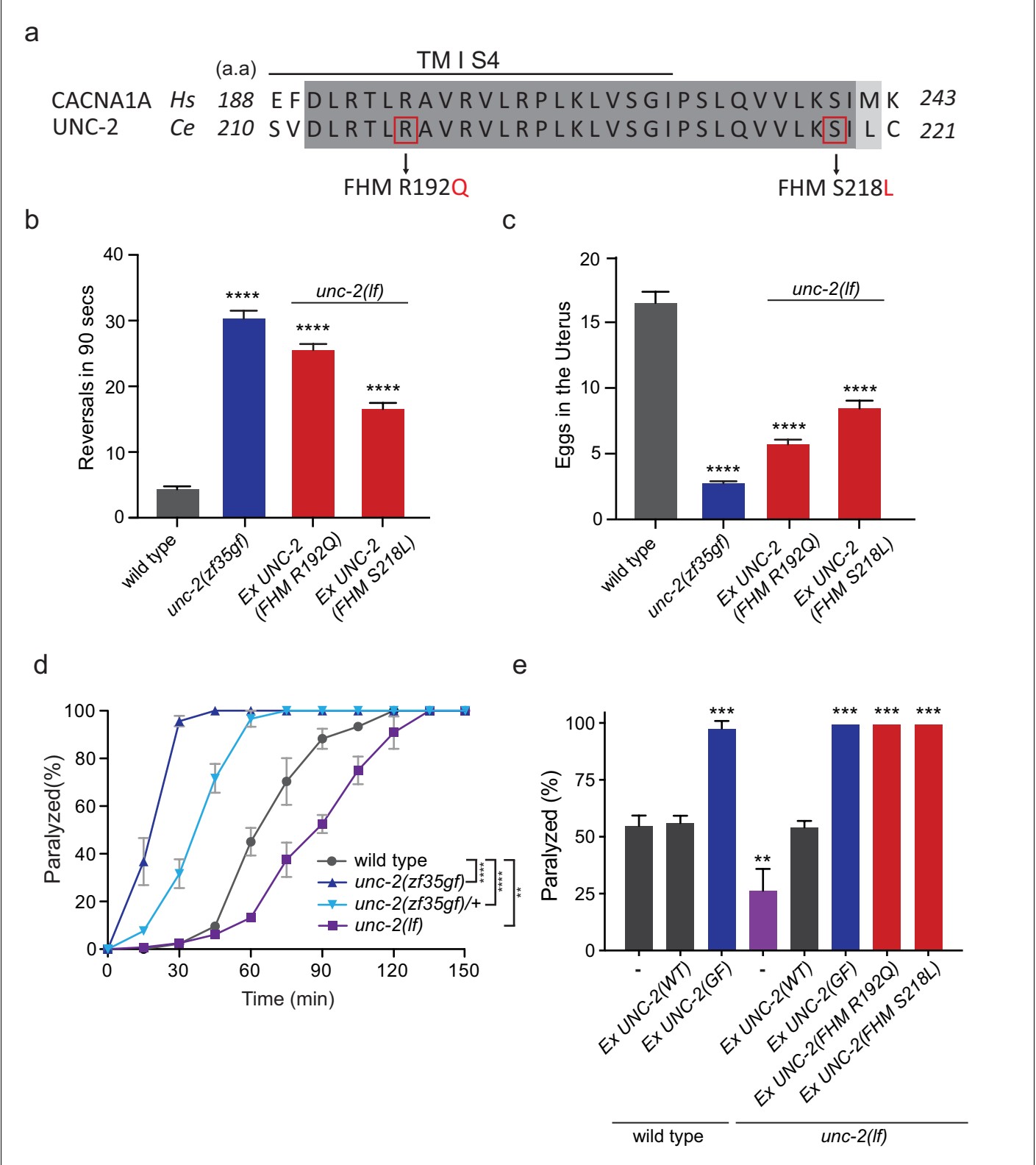

**Figure 4.** FHM1 mutations in *unc-2* gene result in a hyperactive phenotype. (**a**) The amino acid alignment of the conserved region of transmembrane domain I membrane-spanning segments 4 (TM I S4) and the following linker region from human (CACNA1A) and worm (UNC-2) CaV2α subunits. Identities are dark gray and similarities are light gray. Indicated are the known human FHM1 mutations: R192Q and S218L. (**b**) Shown is the average number of reversals in 90 s on thin lawn OP50 plates: wild type (4.2 ± 0.5, n = 29), *unc-2(zf35gf)* (30.3 ± 1.2, n = 20), *Ptag-168::UNC-2(R192Q)* (25.5 ± 0.9,

*Figure 4 continued on next page*

Figure 4 continued

n = 34), and *Ptag-168::UNC-2(R192Q)* (16.5 ± 0.9, n = 33). (c) Average numbers of eggs in the adult uterus: wild type (16.5 ± 0.8 eggs, n = 23), *unc-2 (zf35gf)* (2.7 ± 0.2, n = 35), *Ptag-168::UNC-2(R192Q)* (5.7 ± 0.4, n = 37), and *Ptag-168::UNC-2(S218L)* (8.4 ± 0.6, n = 32). Each bar represents the mean ± SEM for at least three trials with indicated totaling animals number. Statistical difference from wild-type, ****p<0.0001, one-way ANOVA with Dunnett's multiple comparisons test. (d) Quantification of paralysis on 1 mM aldicarb. Each data point represents the mean ± SEM of the percentage of animals paralyzed every 15 min. 50% of the wild-type animals were paralyzed at 60 min. *unc-2(lf)* animals were resistant to the effects of aldicarb and reached 50% paralysis at 90 min. Homozygous *unc-2(zf35gf)* mutants were sensitive to aldicarb; 50% of the *unc-2(zf35gf)* mutants were paralyzed at 20 min. 50% of heterozygous *unc-2(zf35gf)* mutants paralyzed at 40 min. Three independent trials with at least 50 animals for each genotype; **p<0.01, ****p<0.0001, two-way ANOVA with Tukey's multiple comparisons test. (e) Quantification of paralysis percentage on 1 mM aldicarb at the 60 min time point: 55.5% ± 4.5 of wild type, 56.7% ± 3.3 of *Ptag-168::UNC-2(WT)* and 98.3% ± 3.3 of *Ptag-168::UNC-2(GF)* expressed in wild-type animals, 27.1% ± 7.3 of *unc-2(lf)* animals, 54.8% ± 2.9 of *Ptag-168::UNC-2(WT)*, 100% of *Ptag-168::UNC-2(GF)*, and 100% of *Ptag-168::UNC-2(R192Q)* and *Ptag-168::UNC-2(S218L)* in *unc-2(lf)* background. **p<0.01, ***p<0.001, one-way ANOVA with Dunnett's multiple comparisons test.
DOI: https://doi.org/10.7554/eLife.45905.010

The following source data is available for figure 4:

**Source data 1.** Source data for *Figure 4*.
DOI: https://doi.org/10.7554/eLife.45905.011

## *unc-2(zf35gf)* mutants exhibit increased cholinergic and decreased GABAergic spontaneous postsynaptic currents (sPSCs) at the neuromuscular junction

To directly assay the effect of the *unc-2(zf35gf)* mutation on synaptic function, we measured the frequency of spontaneous neurotransmitter release events in recordings of postsynaptic currents (PSCs) from *C. elegans* body wall muscles. *C. elegans* body wall muscles are innervated by both excitatory (cholinergic) and inhibitory (GABAergic) motor neurons (*White et al., 1986*; *McIntire et al., 1993*; *Lewis et al., 1980*), To examine the total spontaneous PSC events, we performed recordings under conditions where both cholinergic and GABAergic PSCs appear as inward currents (−60 mV holding potential, see Material and methods). *unc-2(zf35gf)* mutants showed an over two-fold increase in the overall frequency of spontaneous PSCs when compared to wild-type animals (*Figure 5A and B*), with no significant changes in the mean amplitude (*Figure 5A and C*).

Since excitatory and inhibitory neurotransmitter systems appear to be differentially affected in FHM1 mouse models (*Tottene et al., 2009*; *Vecchia et al., 2014*; *Vecchia et al., 2015*), we analyzed the effect of the *unc-2(zf35gf)* mutation on cholinergic and GABAergic transmission. To isolate cholinergic currents, we performed recordings at a holding potential of −60 mV in the GABA receptor/ *unc-49* mutant background. The frequency of spontaneous excitatory postsynaptic currents (EPSCs) was increased by approximately 1.5-fold in *unc-2(zf35gf); unc-49* double mutants compared with control *unc-49* single mutants (*Figure 5D–F*). To isolate spontaneous GABAergic inhibitory postsynaptic currents (IPSCs), we performed recordings in the presence of 0.5 mM d-tubocurarine at a holding potential of −10 mV, a condition that specifically eliminates EPSCs (*Maro et al., 2015*). The frequency of spontaneous IPSC was reduced by half, without significant changes in the amplitude (*Figure 5G–I*).

Our data show that, despite being expressed by both cholinergic and GABAergic motor neurons, the *unc-2(zf35gf)* mutation leads to increased cholinergic and decreased GABAergic transmission to body wall muscles. Thus, instead of causing a uniform increase of neural signaling, the UNC-2/ CaV2α(GF) mutation differentially affects excitatory and inhibitory signaling, shifting the E/I balance toward excitatory transmission.

## *unc-2(zf35gf)* differentially affects excitatory and inhibitory synapses

How does an increase of UNC-2/CaV2α activity lead to an E/I imbalance? Since changes in neuronal activity can modulate synaptic protein distribution (*Frank, 2014*; *Turrigiano, 2012*), we examined the morphology of pre- and post-synaptic markers at cholinergic and GABAergic NMJs (*Figure 6*). We labeled cholinergic NMJs with the presynaptic vesicle marker RAB-3::mCherry (*Pacr-2*::RAB-3:: mCherry) and the postsynaptic nicotinic ACh receptor (AChR) UNC-29::GFP (*Punc-29*::UNC-29::GFP) (*Figure 6A–D*). RAB-3::mCherry puncta were larger in *unc-2(zf35gf)* mutants (*Figure 6A and B*), consistent with the notion that increased calcium influx can recruit more synaptic vesicles to release sites (*Thanawala and Regehr, 2013*). Importantly, we also observed a marked increase in the size of

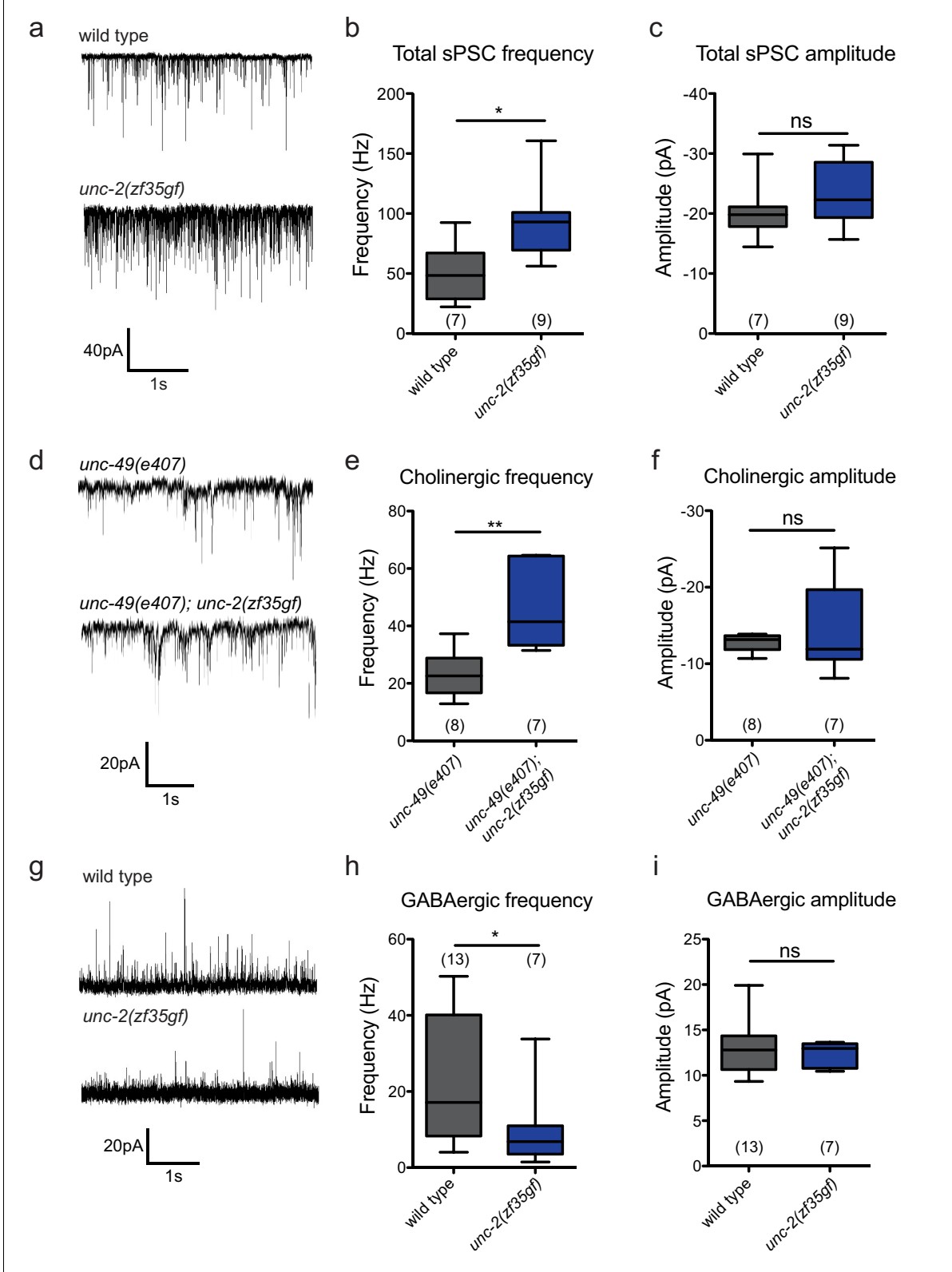

**Figure 5.** The *unc-2(zf35gf)* mutation leads to increased spontaneous EPSCs and decreased spontaneous IPSCs. (**a**) Representative traces of total spontaneous postsynaptic currents (sPSCs) from ventral body wall muscles in wild-type and *unc-2(zf35gf)* mutants. (**b and c**) Mean spontaneous PSC frequency and amplitude of wild-type and *unc-2(zf35gf)* mutants. (**d**) Representative traces of spontaneous cholinergic EPSCs in *unc-49* and *unc-49; unc-2(zf35gf)* mutants. (**e and f**) Mean spontaneous EPSC frequency and amplitude *unc-49* and *unc-49; unc-2(zf35gf)* mutants. (**g**) Representative traces of

*Figure 5 continued on next page*

*Figure 5 continued*

spontaneous GABAergic IPSCs in wild-type and *unc-2(zf35gf)* mutants. (**h and i**) Mean IPSC frequency and amplitude of wild-type animals and *unc-2 (zf35gf)* mutants. Error bars depict SEM. *p<0.05, **p<0.01, two-tailed Student's t test.

DOI: https://doi.org/10.7554/eLife.45905.012

The following source data is available for figure 5:

**Source data 1.** Source data for *Figure 5*.

DOI: https://doi.org/10.7554/eLife.45905.013

UNC-29::GFP clusters, indicating a concomitant increase in the postsynaptic receptors (*Figure 6C and D*). To pharmacologically test if the increase in UNC-29::GFP fluorescence reflects an increase in the expression of functional AChRs at the cell surface, we examined the response of *unc-2(zf35gf)* mutants to an AChR agonist, levamisole. Levamisole induces hyper-contraction and paralysis through the activation of a class of UNC-29-containing AChRs in body wall muscles (*Lewis et al., 1980*). *unc-2(zf35gf)* mutants were hypersensitive to levamisole, consistent with an increased AChR expression on the muscle cell membrane (*Figure 6—figure supplement 1*). These pre- and postsynaptic morphological changes and pharmacological responses are consistent with the notion that the UNC-2/CaV2α(GF) mutation increases excitatory signaling to body wall muscle cells.

We observed a different effect on GABAergic NMJ morphology. We visualized GABAergic NMJs with the same presynaptic vesicle marker RAB-3 (*Punc-25*::RAB-3::mCherry) and the GABA$_A$ receptor UNC-49 (*Punc-49*::UNC-49::GFP). In *unc-2(zf35gf)* mutants, RAB-3::mCherry puncta were enlarged, to a level comparable to that observed for cholinergic NMJs (*Figure 6E and F*). However, UNC-49::GFP puncta were severely reduced in both size and number (*Figure 6G and H*). In *unc-2(zf35gf)* mutants, RAB-3::mCherry puncta density was slightly increased, whereas UNC-49::GFP puncta density was reduced compared to wild type (*Figure 6—figure supplement 2*). At some NMJs, we noted the presence of RAB-3::mCherry puncta without punctate UNC-49::GFP apposition (*Figure 6—figure supplement 3*), suggesting post-synaptic silencing of GABA synapses.

The reduced UNC-49::GFP fluorescence in *unc-2(zf35gf)* mutants is in sharp contrast to the increased fluorescence of the UNC-29::GFP cholinergic receptor. RNA-seq experiments showed no obvious changes in the *unc-49* and *unc-29* expression level in wild-type *vs unc-2(zf35gf)* animals (*Figure 6—figure supplement 4*), suggesting post-transcriptional changes in UNC-49 and UNC-29 receptor localization and distribution. To determine if the morphological changes in UNC-49::GFP fluorescence signals reflect reduced levels of functional UNC-49 on the muscle cell surface, we analyzed *unc-2(zf35gf)* mutants' response to the GABA receptor agonist muscimol. Muscimol induces hyperpolarization of body wall muscles through UNC-49/GABA$_A$-mediated inward Cl$^-$ currents (*Richmond and Jorgensen, 1999*). Muscimol sensitivity is assessed by the animal's ability to respond to head touch. Wild-type animals typically initiate backward locomotion when touched to their heads. After treatment with 1 mM muscimol, severely affected wild-type animals become flaccid, unable to respond to head touch. Moderately affected animals respond with a rubber band phenotype, in which the body wall muscles initially contract but then fully relax, failing to generate backward locomotion (*de la Cruz et al., 2003*). *unc-2(zf35gf)* mutants exhibited reduced sensitivity to muscimol: most *unc-2(zf35gf)* mutants were able to generate backward locomotion upon the head touch (*Figure 6—figure supplement 1B*). The partial resistance of *unc-2(zf35gf)* mutants to muscimol-induced muscle relaxation is consistent with reduction of UNC-49/GABA$_A$ expression at the muscle cell surface.

Thus, consistent with the electrophysiological analyses, our pharmacological studies demonstrate that a gain-of-function mutation in UNC-2/CaV2α has distinct effects on cholinergic and GABAergic synapses. However, both spontaneous EPSC and IPSC amplitudes are not significantly different between *unc-2(zf35gf)* mutants and the wild type. This suggests that the density of functional cholinergic and GABAergic receptors at individual synapses is unchanged in *unc-2(zf35gf)* mutants. As individual synapses can be difficult to resolve with confocal microscopy, single fluorescent puncta often represent multiple synapses. Therefore, the synaptic fluorescence changes we observe most likely indicate increases or decreases in the number of excitatory and inhibitory synaptic connections. An increased number of cholinergic synapses would account for the increased sEPSC frequency, the levamisole hypersensitivity, and increased UNC-29::GFP fluorescence intensity in the of nerve cord.

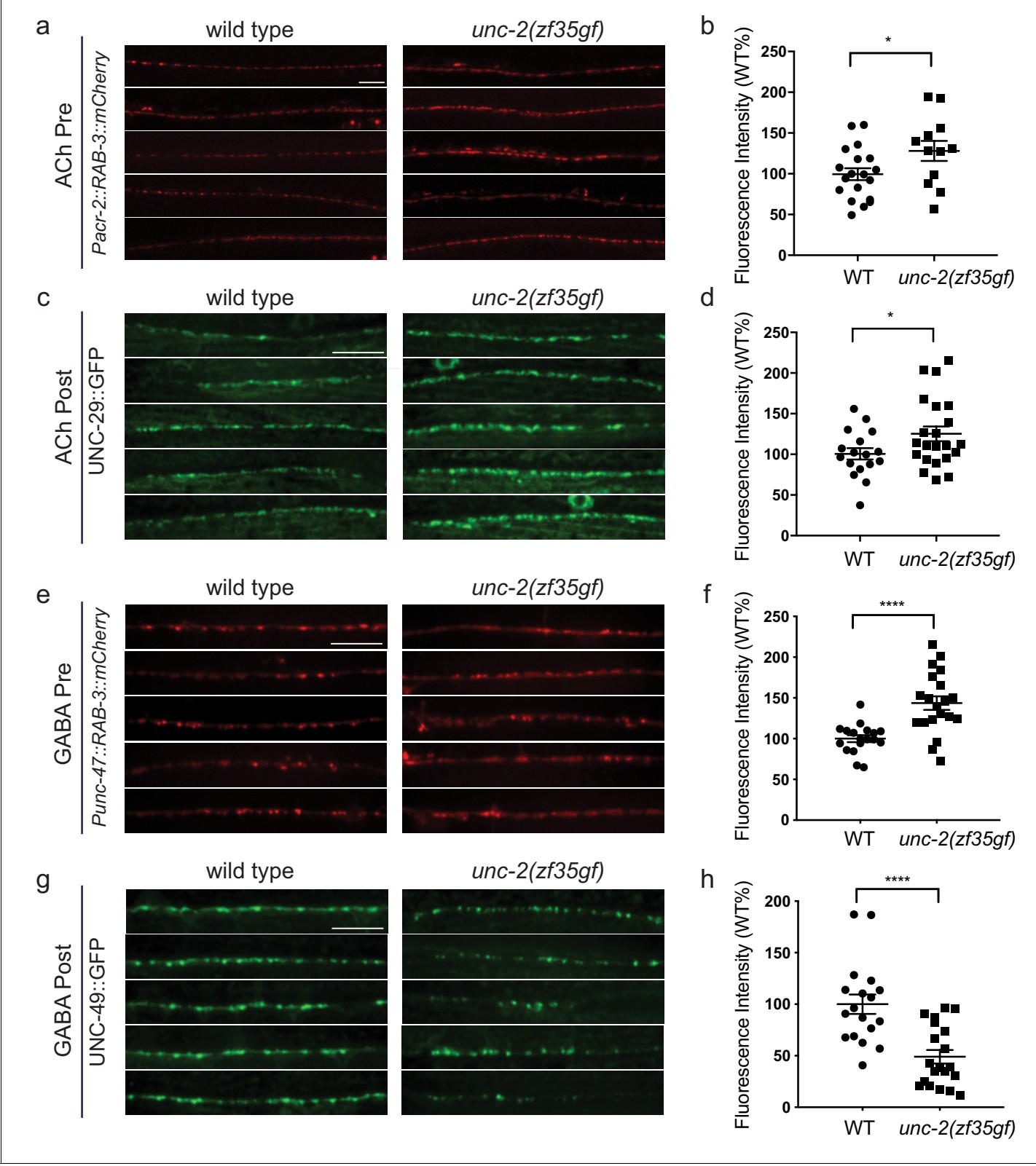

**Figure 6.** *unc-2(zf35gf)* mutants have decreased GABA_A receptor expression at the NMJ. (**a and c**) Representative images of cholinergic synapses in wild type and *unc-2(zf35gf)* mutants. Presynaptic sites are labeled with synaptic vesicle marker RAB-3::mCherry while postsynaptic nicotinic acetylcholine receptors are labeled by UNC-29::GFP. Scale bar represents 10 µm. (**b and d**) Quantification of the fluorescence intensity of RAB-3::mCherry and UNC-29::GFP along the ventral nerve cord at cholinergic synapses in wild type and *unc-2(zf35gf)* animals. Arbitrary fluorescence units of individual animals are

*Figure 6 continued on next page*

*Figure 6 continued*

normalized to the mean value of the wild type. Normalized fluorescence of cholinergic RAB-3::mCherry: 0.99 ± 0.76, n = 21 in wild type and 1.36 ± 0.18, n = 14 in *unc-2(zf35gf)* mutants. UNC-29::GFP: 0.97 ± 0.05, n = 35 in wild type and 1.19 ± 0.06, n = 38 in *unc-2(zf35gf)* mutants. (**e** and **g**) Representative images of GABAergic synapses in wild type and *unc-2(zf35gf)* mutants. Presynaptic sites are labeled with synaptic vesicle marker RAB- 3::mCherry while postsynaptic GABA receptors are labeled by UNC-49::GFP. Scale bar represents 10 μm. (**f** and **h**) Quantification of the fluorescence intensity of RAB-3::mCherry and UNC-49::GFP along the ventral nerve cord at GABAergic synapses in wild-type and *unc-2(zf35gf)* animals. Arbitrary fluorescence units of individual animals are normalized to the mean value of the wild type. Normalized fluorescence of GABAergic RAB-3::mCherry: 1 ± 0.07, n = 18 in wild type and 1.25 ± 0.08, n = 20 in *unc-2(zf35gf)* mutants. UNC-49::GFP: 1 ± 0.09, n = 18 in wild-type and 0.75 ± 0.06, n = 20 in *unc-2(zf35gf)* animals. For all the quantification above, error bars depict SEM. *p<0.05, ****p<0.0001, two-tailed Student's t test.

DOI: https://doi.org/10.7554/eLife.45905.014

The following source data and figure supplements are available for figure 6:

**Source data 1.** Source data for *Figure 6*.
DOI: https://doi.org/10.7554/eLife.45905.022
**Figure supplement 1.** *unc-2(zf35gf)* mutants are hypersensitive to the AChR agonist, levamisole and resistant to the GABA receptor agonist, muscimol.
DOI: https://doi.org/10.7554/eLife.45905.015
**Figure supplement 1—source data 1.** Source data for *Figure 6—figure supplement 1*.
DOI: https://doi.org/10.7554/eLife.45905.016
**Figure supplement 2.** *unc-2(zf35gf)* mutants have an increased RAB-3 puncta density and a reduced UNC-49 puncta density along the nerve cord.
DOI: https://doi.org/10.7554/eLife.45905.017
**Figure supplement 2—source data 1.** Source data for *Figure 6—figure supplement 2*.
DOI: https://doi.org/10.7554/eLife.45905.018
**Figure supplement 3.** Representative images of GABAergic synapses pre- and post-synaptic apposition in wild-type and *unc-2(zf35gf)* animals.
DOI: https://doi.org/10.7554/eLife.45905.019
**Figure supplement 4.** wild-type and *unc-2(zf35gf)* animals have similar *unc-29* and *unc-49* mRNA levels.
DOI: https://doi.org/10.7554/eLife.45905.020
**Figure supplement 4—source data 1.** Source data for *Figure 6—figure supplement 4*.
DOI: https://doi.org/10.7554/eLife.45905.021

Similarly, a reduced number of GABAergic synapses is consistent with a reduced spontaneous IPSC frequency, reduced sensitivity to muscimol, and reduced UNC-49::GFP fluorescence intensity in the nerve cord.

## *unc-2(zf35gf)* expression in cholinergic neurons impairs GABA synapse formation

The striking difference in excitatory and inhibitory neuromuscular signaling in *unc-2(zf35gf)* mutants is surprising since both cholinergic and GABAergic neurons express *unc-2*. Why does a gain-function-mutation in the presynaptic CaV2 channel lead to a reduction in the number of GABAergic synapses? The simplest explanation is that cholinergic and GABAergic synapses respond differently to increased presynaptic activity. For instance, while increased ACh release may result in the increase of cholinergic synapses, increased GABA release may result in a homeostatic reduction of GABAergic synapses. To test this possibility, we analyzed the GABAergic synaptic markers in animals that specifically express the *unc-2(zf35gf)* transgene in GABAergic motor neurons (*Punc-47*::UNC-2(GF)) in a wild-type background. Expression in GABAergic motor neurons alone resulted in an increase in both presynaptic RAB-3::mCherry and post-synaptic UNC-49::GFP fluorescence (*Figure 7A and B*; *Figure 7—figure supplement 1*). *Punc-47*::UNC-2(GF) animals are partially resistant to aldicarb, indicative of increased GABAergic signaling onto the body wall muscles (*Figure 7—figure supplement 2*). Thus, the reduction of GABAergic synapses in *unc-2(zf35gf)* mutants is not a direct consequence of elevated GABAergic neuron activity. Instead, our results indicate that increased GABAergic motor neuron activity in principle leads to increases in both presynaptic and postsynaptic termini similar to that observed for cholinergic synapses.

Previous studies showed that cholinergic signaling affects the development and transmission of GABAergic neurons (*Jospin et al., 2009*; *Barbagallo et al., 2017*). Therefore, increased cholinergic transmission in *unc-2(zf35gf)* mutants may negatively affect the formation of GABAergic synapses. Indeed, when we expressed *unc-2(zf35gf)* only in cholinergic neurons (*Pacr-2*::UNC-2(GF)), UNC-49::GFP fluorescence was reduced to a similar degree as in the *unc-2(zf35gf)* mutants (*Figure 7A and*

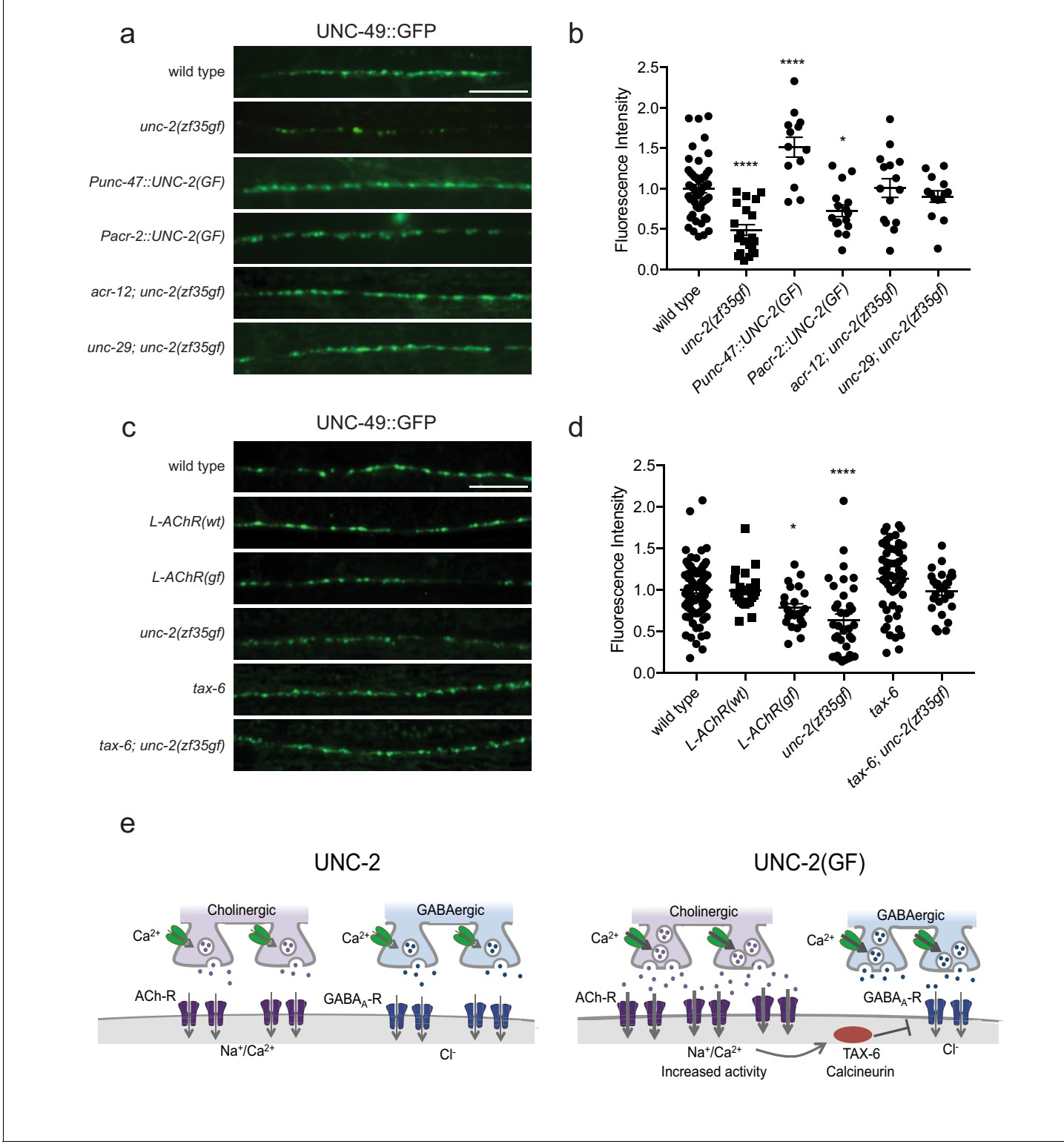

**Figure 7.** The reduction of GABA_A receptor in *unc-2(zf35gf)* mutants is dependent on nicotinic acetylcholine receptor mediated signaling. (**a and c**) Representative images of GABAergic post-synaptic sites labeled with UNC-49::GFP of indicated genotypes. Scale bar represents 10 μm (**b**) Quantification of the fluorescence intensity of UNC-49::GFP along the nerve cord. Arbitrary fluorescence units of individual animals are normalized to the mean value of wild type. Normalized UNC-49::GFP fluorescence: wild type (1 ± 0.06, n = 41), *unc-2(zf35gf)* (0.4 ± 0.07, n = 15), *Punc-47::UNC-2(GF)* (1.5 ± 0.12, n = 13), *Pacr-2::UNC-2(GF)* (0.7 ± 0.07, n = 16), *acr-12; unc-2(zf35gf)* (1 ± 0.11, n = 15) and *unc-29; unc-2(zf35gf)* (0.9 ± 0.07, n = 14). (**d**) Quantification of the fluorescence intensity of UNC-49::GFP along the nerve cord of indicated genotypes. Arbitrary fluorescence units of individual

*Figure 7 continued on next page*

*Figure 7 continued*

animals are normalized to the mean value of wild type. Normalized UNC-49::GFP fluorescence: wild type (1 ± 0.04, n = 82), *L-AChR(WT)* (1 ± 0.05, n = 22), *L-AChR(GF)* (0.8 ± 0.05, n = 23), *unc-2(zf35gf)* (0.6 ± 0.07, n = 36), *tax-6* (1.1 ± 0.06, n = 54), *tax-6; unc-2(zf35gf)* (1 ± 0.05, n = 29). For all the quantification above, error bars depict SEM. *$p<0.05$, ****$p<0.0001$, one-way ANOVA with Dunnett's multiple comparisons. (**e**) Model: The UNC-2 gain-of-function mutation shifts the E/I balance to an excitation-dominant transmission through the destabilztion of GABA synapses in a TAX-6/calcineurin-dependent manner (See text for explanation).

DOI: https://doi.org/10.7554/eLife.45905.023

The following source data and figure supplements are available for figure 7:

**Source data 1.** Source data for *Figure 7*.

DOI: https://doi.org/10.7554/eLife.45905.030

**Figure supplement 1.** Quantification of RAB-3 vesicle marker in GABAergic synapses.

DOI: https://doi.org/10.7554/eLife.45905.024

**Figure supplement 1—source data 1.** Source data for *Figure 7—figure supplement 1*.

DOI: https://doi.org/10.7554/eLife.45905.025

**Figure supplement 2.** Cell-specific expression of *unc-2(zf35gf)* transgene in cholinergic or GABAergic motor neurons confers corresponding aldicarb response.

DOI: https://doi.org/10.7554/eLife.45905.026

**Figure supplement 2—source data 1.** Source data for *Figure 7—figure supplement 2*.

DOI: https://doi.org/10.7554/eLife.45905.027

**Figure supplement 3.** Knocking down *tax-6* gene expression in non-neuonal cells is sufficient to suppress the reduction of UNC-49::GFP in *unc-2 (zf35gf)* mutants.

DOI: https://doi.org/10.7554/eLife.45905.028

**Figure supplement 3—source data 1.** Source data for *Figure 7—figure supplement 3*.

DOI: https://doi.org/10.7554/eLife.45905.029

*B*). Presynaptic RAB-3::mCherry fluorescence in GABAergic neurons was slightly increased in *Pacr-2*::UNC-2(GF) animals (*Figure 7—figure supplement 1*), which may reflect increased stimulation of GABAergic motor neurons by cholinergic motor neurons. Expression of an *unc-2(zf35gf)* transgene in cholinergic motor neurons increased sensitivity to aldicarb, consistent with an expected increase in cholinergic signaling (*Figure 7—figure supplement 2*). Together, these results suggest that increased activity of cholinergic motor neurons in *unc-2(zf35gf)* mutants is not only required but also causes the decrease in GABAergic synapses.

## Increased excitatory signaling leads to calcineurin-dependent reduction of inhibitory synapses

How might increased cholinergic input lead to a reduction in GABAergic synapses? First, we examined whether reducing cholinergic synaptic transmission was sufficient to restore UNC-49::GFP expression in *unc-2(zf35gf)* mutants. ACR-12, expressed by cholinergic motor neurons, and UNC-29, expressed by body wall muscles, are subunits of ionotropic AChRs. The loss of ACR-12 reduces excitability of cholinergic motor neurons (*Jospin et al., 2009*; *Petrash et al., 2013*). Loss of UNC-29, a subunit of the levamisole-sensitive AChR, reduces cholinergic depolarization of body wall muscles (*Fleming et al., 1997*; *Richmond and Jorgensen, 1999*). In both *unc-2(zf35gf); acr-12* and *unc-2 (zf35gf); unc-29* mutants, UNC-49::GFP fluorescence was restored to wild-type levels (*Figure 7A and B*). This finding indicates that increased cholinergic input to body wall muscles is the primary signal for decreasing the number of GABA synapses. Cholinergic motor neurons simultaneously innervate body wall muscles and GABAergic motor neurons (*White et al., 1986*). Both *acr-12* and *unc-29* are also expressed by GABAergic motor neurons, and play a role in cholinergic activation of not only body wall muscles, but also GABA motor neurons (*Petrash et al., 2013*; *Philbrook et al., 2018*). However, GABA signaling is not required for UNC-49/GABA$_A$R expression or localization in body wall muscles (*Gally and Bessereau, 2003*). Together, these results suggest that increased cholinergic input to body wall muscles negatively regulates GABAergic postsynapse formation or stability.

To directly test this possibility, we examined UNC-49::GFP expression in animals where we specifically increased cholinergic input to body wall muscles. Muscle-specific expression of the hyperactive levamisole-sensitive AChR (L-AChR(GF)) containing gain-of-function mutations in L-AChR subunits UNC-29 and UNC-38, leads to increased excitation of body wall muscles, but no obvious defects in

muscle structure or cholinergic synapses (*Bhattacharya et al., 2014*). *L-AChR(gf)* transgenic animals exhibited normal presynaptic marker expression at GABAergic NMJs. However, similar to *unc-2 (zf35gf)* mutants, postsynaptic UNC-49::GFP fluorescence was markedly reduced in *L-AChR(gf)* animals (*Figure 7C and D*). Transgenic expression of the wild-type L-AChR (*Pmyo-3*::L-AChR(wt)) did not affect UNC-49::GFP fluorescence. This indicates that increased cholinergic signaling onto muscles in *unc-2(zf35gf)* mutants negatively regulates GABAergic postsynapse formation.

Several studies with cultured hippocampal neurons suggest GABAergic receptors are modulated by excitatory neuronal activity. In particular, sustained high $Ca^{2+}$ levels reduce inhibitory synaptic strength through a calcineurin-dependent lateral diffusion of $GABA_A$ receptor from synapses (*Bannai et al., 2009*; *Bannai et al., 2015*; *Muir et al., 2010*). We examined whether *C. elegans* calcineurin, TAX-6, is required for the decrease in GABAergic postsynapses in *unc-2(zf35gf)* mutants. UNC-49::GFP expression was not significantly different in *tax-6(lf)* mutants (*Figure 7C and D*). However, the *tax-6(lf)* mutation restored UNC-49::GFP fluorescence in *unc-2(zf35gf)* mutants. *tax-6* is expressed in muscles and several neurons (*Kuhara et al., 2002*). To determine whether *tax-6* is required in muscle, we performed RNAi feeding experiments. Most *C. elegans* neurons are resistant to RNAi feeding (*Kamath et al., 2001*; *Timmons et al., 2001*). *tax-6* RNAi feeding restored UNC-49::GFP fluorescence in *unc-2(zf35gf)* mutants (*Figure 7—figure supplement 3*), suggesting that TAX-6 acts in muscle to regulate GABA synapses. Together, these results indicate that increased cholinergic input to body wall muscles reduces the number of GABAergic postsynapses in a calcineurin-dependent manner.

## Discussion

### Gain- and loss-of-function mutations in *unc-2/CaV2$\alpha$* result in opposing phenotypes

Presynaptic voltage-gated calcium channels (CaV2) are crucial regulators of neuronal excitability and synaptic transmission. Here, we report the isolation of a gain-of-function mutation in the *unc-2* gene, which encodes the CaV2$\alpha$ subunit gene of *C. elegans*. *unc-2(zf35gf)* mutants are hyperactive and exhibit seizure-like motor behaviors, in contrast to the sluggish behavior of *unc-2(lf)* mutants (*Schafer and Kenyon, 1995*; *Mathews et al., 2003*). The *unc-2(zf35gf)* mutation results in a G-to-R substitution in a highly conserved region in the intracellular linker between TMIII and TMIV. Our electrophysiological analyses of the human CACNA1A channel in HEK cells indicate that this G-to-R substitution causes a shift to lower voltages of activation and reduced inactivation of the channel to increase $Ca^{2+}$ influx. The increased current density of the CACNA1A(G1518R) channel could reflect increased channel conductance, and/or enhanced cell surface expression. We did not observe obvious differences in the expression and localization of the UNC-2(WT) and UNC-2(GF/G1132R) channel in *C. elegans.* This could suggest that this G-to-R substitution in the CaV2 channel may arise from an increased channel conductance. However, we cannot exclude that enhanced expression of the CACNA1A(G1518R) channel in HEK cells culture contributes to the increased current density. A similar G-to-R substitution in an intracellular linker of the human CaV1.2 channel results in similar defects in channel inactivation that underlies Timothy syndrome (*Splawski et al., 2004*). The negative shift in the activation potential of UNC-2/CaV2$\alpha$(GF) channel is reminiscent of similar observations for several mutant human CaV2.1$\alpha$ channels that have been identified in patients with familial hemiplegic migraine type 1 (FHM1) (*Hans et al., 1999*; *Tottene et al., 2005*; *Müllner et al., 2004*). While both loss- and gain-of-function phenotypes in CaV2.1 channels with FHM1 mutations have been reported in various expression systems, most FHM1 mutations appear to lead to channel activation at lower voltages and/or increased channel open probability. The gain-of-function effect of FHM1 mutations is supported by knock-in mouse models of the FHM1 R192Q and S218L channel, which activate at lower membrane potentials and have an increase in open probability (*Tottene et al., 2009*; *van den Maagdenberg et al., 2010*).

Intragenic suppressor mutations of the *unc-2(zf35gf)* allele include both premature stop codons and missense mutations. Most intragenic suppressor mutations result in uncoordinated and lethargic phenotypes, indicating that they are hypomorphic alleles. Interestingly, some intragenic suppressor mutations resemble those found in CACNA1A in episodic ataxia type 2 (EA2) patients. The UNC-2 (C341Y) mutation in the domain I S5-S6 loop is analogous to the CACNA1A(C287Y) mutation which

was shown to alter channel trafficking and kinetics in whole-cell patch-clamp recordings of transfected COS-7 cells (*Wan et al., 2005*), The UNC-2(L1355F) mutation in the domain IV S5-S6 loop analogous to CACNA1A (L1749P) mutation which was identified in a genome wide association study of EA2 patients (*Maksemous et al., 2016*). These and other CACNA1A(EA2) missense mutations are partial or total loss-of-function mutations that lead to defects in channel trafficking or positive shifts in the voltage threshold for activation (*Jeng et al., 2008*; *Mezghrani et al., 2008*).

We found that expression of an *unc-2* transgene carrying FHM1 mutations R192Q and S218L in *C. elegans* recapitulated the behavioral hyperactivity of *unc-2(zf35gf)* mutants, whereas EA2-like CACNA1A(lf) mutations led to decreased motor activity. These studies provide strong genetic evidence that EA2 mutations are reduction-of-function mutations, while FHM1 mutations are gain-of-function mutations. *C. elegans*, which has a single CaV2$\alpha$ gene, thus provides an efficient in vivo system to determine the genetic nature of VGCC mutations associated with neurological disorders.

## An *unc-2* gain-of-function mutation results in E/I imbalance

Presynaptic Ca$^{2+}$ influx through CaV2 channels is tightly coupled to neurotransmitter release. *unc-2* loss-of-function mutants are resistant to the acetylcholinesterase inhibitor aldicarb (*Miller et al., 1996*), and have a reduction in spontaneous EPSC frequency (*Richmond et al., 2001*; *Tong et al., 2015*; *Liu et al., 2018*). The *unc-2(zf35gf)* mutation increases Ca$^{2+}$ influx, which would lead to an increase in neurotransmitter release probability. In accordance, *unc-2(zf35gf)* mutants are hypersensitive to aldicarb, and show a two-fold increase in spontaneous EPSC frequency. In contrast, spontaneous IPSC frequency is significantly reduced in *unc-2(zf35gf)* mutants. Therefore, even though UNC-2 is expressed by both cholinergic and GABAergic motor neurons, the UNC-2/CaV2$\alpha$(GF) mutation differentially affects excitatory and inhibitory signaling, shifting the E/I balance toward excitatory transmission.

Human studies indicate that cortical hyperexcitability in migraine patients (*Aurora and Wilkinson, 2007*; *Pietrobon and Striessnig, 2003*), could result from enhanced excitation and/or reduced inhibition. This has led to the hypothesis that migraine is a disorder of brain E/I imbalance (*Vecchia and Pietrobon, 2012*; *Mainero and Louapre, 2014*). Our data strongly support this hypothesis. The differential effect on excitatory and inhibitory signaling was also observed in FHM1 mouse models (*Tottene et al., 2009*). The R192Q FHM1 knock-in mice exhibit increased excitatory glutamatergic signaling, while inhibitory GABAergic transmission appears unaffected. In the R192Q FHM1 mice, an increase in glutamate release is thought to play a key role in initiation of cortical spreading depression, but the molecular and cellular mechanisms that underlie this E/I imbalance in mammals remain unclear. Our results provide new insights into how CaV2 gain-of-function mutations may lead to the E/I imbalances.

## Increased excitatory transmission leads to destabilization of GABAergic synapses

The *C. elegans* neuromuscular system, where both excitatory (cholinergic) and inhibitory (GABAergic) motor neurons regulate muscle activity, provides a suitable and complementary model for mechanistic studies of E/I imbalance (*Stawicki et al., 2011*; *Safdie et al., 2016*; *Zhou et al., 2017*). In our system, *unc-2(zf35gf)* mutations led to a modest increase in RAB-3 expression in the neurites of both excitatory and inhibitory motor neurons, consistent with the notion that increased Ca$^{2+}$ influx may potentiate the recruitment of synaptic vesicles (*Gracheva et al., 2008*; *Han et al., 2011*). However, the *unc-2(zf35gf)* mutation led to pronounced and opposite effects on the density of cholinergic and GABAergic receptors in the ventral nerve cord: an increase of AChR, but a marked decrease of GABA$_A$R, which parallel the increased sEPSC frequency and reduced sIPSC frequency. Like the wild-type UNC-2/CaV2$\alpha$, UNC-2/CaV2$\alpha$(GF) channel proteins localize to presynapses to mediate Ca$^{2+}$ influx and exocytosis of neurotransmitters so these effects are not attributable to channel mislocalization. Unchanged amplitudes of spontaneous EPSCs and IPSCs suggest the density of receptors at individual cholinergic and inhibitory synapse is not affected by *unc-2(zf35gf)*. Therefore, the differential density of cholinergic and GABAergic receptors and the frequency of spontaneous EPSCs and IPSCs likely reflects the number of functional synapses from excitatory and inhibitory motor neurons to the body wall muscles.

Our results show that the reduced GABAergic neuromuscular signaling in *unc-2(zf35gf)* mutants is a consequence of increased cholinergic signaling onto the same muscle target. A possible explanation of this observation is a differential response of cholinergic and GABAergic synapses to increased stimulation: strengthening of excitatory and homeostatic compensation of inhibitory synapses (*Malenka and Bear, 2004*; *Glanzman, 2010*; *Gaiarsa et al., 2002*). However, our results argue against this possibility: GABAergic-specific expression of the gain-of-function UNC-2/CaV2α channel leads to increased density of GABA$_A$ receptors on the muscles, hence the CaV2(GF) channel in principle should increase synaptic strength in both synapse types. Our results instead reveal that the reduction of GABAergic neuromuscular signaling is a consequence of increased cholinergic input to the muscle cells.

In the mammalian brain, excessive neuronal excitation can induce long-term depression of GABAergic transmission (*Gaiarsa et al., 2002*). Long-term depression of GABAergic transmission is associated with decreased GABA$_A$ receptor clustering (*Bannai et al., 2009*). NMDA receptor mediated Ca$^{2+}$ influx can induce LTD at GABAergic synapses by activating calcineurin (*Lu et al., 2000*; *Wang et al., 2003*). Sustained activity-dependent Ca$^{2+}$ influx reduces inhibitory synaptic strength through a calcineurin-dependent increase in the lateral mobility of synaptic GABA$_A$ receptors (*Bannai et al., 2009*; *Muir et al., 2010*). GABA$_A$ receptor clustering is regulated in part by lateral diffusion on the cell surface (*Triller and Choquet, 2008*), utilizing several evolutionarily conserved molecular mechanisms (*Maro et al., 2015*; *Tong et al., 2015*; *Tu et al., 2015*). In *C. elegans*, increased AChR-mediated Na$^+$/Ca$^{2+}$ influx in *unc-2(zf35gf)* mutants may similarly affect GABA$_A$ receptor stability to disassemble or prevent the assembly of GABAergic post-synapses. Increased excitatory signaling may thus lead to silencing of GABA synapses at the postsynaptic site, UNC-49/GABA$_A$ receptor localization to postsynaptic sites is restored by removing TAX-6/calcineurin, implicating a conserved activity-dependent mechanism for modulation of synaptic inhibition.

We propose that UNC-2/CaV2 gain-of-function mutations change the E/I balance of the *C. elegans* neuromuscular system: increased excitatory signaling strengthens excitatory synapses, resulting in the destabilization of inhibitory synapses in a calcineurin-dependent manner *Figure 7E*. A decrease in synaptic inhibition has been implicated in epilepsy, schizophrenia and autism (*Eichler and Meier, 2008*; *Nelson and Valakh, 2015*; *Vecchia and Pietrobon, 2012*). Since the role of CaV2 channels in excitatory and inhibitory signaling is conserved, the processes we describe provide valuable insights into molecular and neural mechanisms of E/I imbalance that underlie neurological disorders.

## Materials and methods

### Strains

All strains were cultured at room temperature (22–24°C) on nematode growth media (NGM) agar plates with the *E. coli* strain OP50 as a food source. Experiments were performed on young adult animals (24 hr post-L4 larva) at room temperature (22–24°C). The wild-type strain was Bristol N2. Transgenic strains were obtained by microinjection of plasmid DNA into the germline with coinjection marker *lin-15* rescuing plasmid pL15EK both at 80 ng/μl into *unc-2(e55); lin-15(n765ts)* or *lin-15 (n765ts)* animals unless stated otherwise. At least three independent transgenic lines were obtained for each injected construct. The data presented are from a single representative line. The following strains were utilized in this study:

### Strains used in this study

| Strain | Feature | Genotype | Figures |
|--------|---------|----------|---------|
| CB55 | canonical *unc-2* loss-of-function | *unc-2(e55)* | *Figure 2*, *Figure 4* |
| QW37 | gain-of-function *unc-2* | *unc-2(zf35gf)* | All Figures |
| QW355 | *unc-2(zf35gf)* intragenic suppressor | *unc-2(zf35gf zf109)* | *Figure 2* |

*Continued on next page*

*Continued*

| Strain | Feature | Genotype | Figures |
|---|---|---|---|
| QW359 | *unc-2(zf35gf)* intragenic suppressor | *unc-2(zf35gf zf113)* | **Figure 2** |
| QW360 | *unc-2(zf35gf)* intragenic suppressor | *unc-2(zf35gf zf114)* | **Figure 2** |
| QW441 | *unc-2(zf35gf)* intragenic suppressor | *unc-2(zf35gf zf115)* | **Figure 2** |
| QW720 | *unc-2(zf35gf)* intragenic suppressor | *unc-2(zf35gf zf124)* | **Figure 2** |
| QW726 | *unc-2(zf35gf)* intragenic suppressor | *unc-2(zf35gf zf130)* | **Figure 2** |
| QW849 | *unc-2(zf35gf)* intragenic suppressor | *unc-2(zf35gf zf134)* | **Figure 2** |
| QW383 | Pan-neuronal expression of UNC-2(GF) | *lin-15(n765ts); zfEx51[Ptag-168::UNC-2(GF); lin-15(+)]* | **Figure 2**, **Figure 4** |
| QW388 | Pan-neuronal expression of UNC-2(GF) in *unc-2(lf)* background | *unc-2(e55); lin-15(n765ts); zfEx51[Ptag-168::UNC-2(GF); lin-15(+)]* | **Figure 2**, **Figure 4** |
| QW392 | Pan-neuronal expression of UNC-2(WT) in *unc-2(lf)* background | *unc-2(e55); lin-15(n765ts); zfEx51[Ptag-168::UNC-2(WT); lin-15(+)]* | **Figure 2**, **Figure 4** |
| QW1632 | Cell-specific expression of expression of UNC-2(GF) in GABAergic motor neurons | *lin-15(n765ts); zfEx801[Punc-47::UNC-2(GF); lin-15(+)]* | **Figure 7—figure supplement 2** |
| QW741 | Cell-specific expression of expression of UNC-2(GF) in cholinergic motor neurons | *lin-15(n765ts); zfEx801[Pacr-2::UNC-2(GF); lin-15(+)]* | **Figure 7—figure supplement 2** |
| QW863 | Pan-neuronal expression of UNC-2(FHM S218L) in *unc-2(lf)* background | *unc-2(e55); lin-15(n765ts); zfEx51[Ptag-168::UNC-2 (FHM1 S218L); lin-15(+)]* | **Figure 4** |
| QW864 | Pan-neuronal expression of UNC-2(FHM R192Q) in *unc-2(lf)* background | *unc-2(lj1); lin-15(n765ts); zfEx51[Ptag-168::UNC-2 (FHM1 R192Q); lin-15(+)]* | **Figure 4** |
| QW1317 | Pan-neuronal expression of UNC-2(WT)::GFP in *unc-2(lf)* background | *unc-2(e55); lin-15(n765ts); zfEx51[Ptag-168::UNC-2(WT)::GFP; lin-15(+)]* | **Figure 4** |
| QW1362 | Pan-neuronal expression of UNC-2(GF)::GFP in *unc-2(lf)* background | *unc-2(e55); lin-15(n765ts); zfEx51[Ptag-168::UNC-2(GF)GFP; lin-15(+)]* | **Figure 4** |
| IZ930 | Synaptic marker strain for GABAergic synapses | *ufIs58[Punc-47::RAB-3::mCherry]; oxIs19[Punc-49::UNC-49::GFP]* | **Figure 6**, **Figure 7**, **Figure 6—figure supplements 2** and **3**, **Figure 7—figure supplements 1** and **3** |
| IZ106 | Synaptic marker strain for nicotinic receptor | *unc-29(x29); ufIs7[Punc-29::UNC-29::GFP]* | **Figure 6** |
| QW937 | Synaptic marker strain for GABAergic synapses in *unc-2(zf35gf)* background | *unc-2(zf35gf); ufIs58[Punc-47::RAB-3::mCherry]; oxIs19[Punc-49::UNC-49::GFP]* | **Figure 6**, **Figure 7**, **Figure 6—figure supplements 2** and **3**, **Figure 7—figure supplements 1** and **3** |
| QW1703 | Synaptic marker strain for GABAergic synapses *in unc-2(zf35gf);acr-12(lf)* background | *unc-2(zf35gf); acr-12(ok367); ufIs58[Punc-47::RAB-3::mCherry]; oxIs19[Punc-49::UNC-49::GFP]* | **Figure 7**, **Figure 7—figure supplement 1** |

*Continued*

| Strain | Feature | Genotype | Figures |
|--------|---------|----------|---------|
| QW1726 | Synaptic marker strain for GABAergic synapses in unc-2(zf35gf);unc-29(lf) background | unc-2(zf35gf); unc-29(x29); ufIs58[Punc-47::RAB-3::mCherry]; oxIs19[Punc-49::UNC-49::GFP] | *Figure 7*, *Figure 7— figure supplement 1* |
| QW1367 | Synaptic marker strain for GABAergic synapses with cell-specific expression of UNC-2(GF) in GABAergic neurons | ufIs58[Punc-47::RAB-3::mCherry]; oxIs19[Punc-49::UNC-49::GFP]; zfEx609[Punc-47::unc-2(zf35gf); +rol-6(+)] | *Figure 7*, *Figure 7— figure supplement 1* |
| QW1375 | Synaptic marker strain for GABAergic synapses with cell-specific expression of UNC-2(GF) in cholinergic motor neurons | ufIs58[Punc-47::RAB-3::mCherry]; oxIs19[Punc-49::UNC-49::GFP]; zfEx613[Pacr-2::unc-2(zf35gf); rol-6(+)] | *Figure 7*, *Figure 7— figure supplement 1* |
| QW1849 | Synaptic marker strain for GABA$_A$ receptor in tax-6(lf) background | tax-6(p675); oxIs19[Punc-49::UNC-49::GFP] | *Figure 7* |
| QW1841 | Synaptic marker strain for GABA$_A$ receptor in unc-2(zf35gf);tax-6(lf) background | unc-2(zf35gf); tax-6(p675); oxIs19[Punc-49::UNC-49::GFP] | *Figure 7* |
| IZ539 | Synaptic marker strain for GABAergic synapses with body wall muscle expression of AChR(GF) | akIs26[Pmyo-3::LEV-1(GF);Pmyo-3::UNC-29(GF); lin-15(+)(L-AChR(GF)]; ufIs58[Punc-47::RAB-3::mCherry]; oxIs19[Punc-49::UNC-49::GFP] | *Figure 7* |
| IZ818 | Synaptic marker strain for GABAergic synapses with body wall muscle expression of AChR(WT) | ufIs47[Pmyo-3::UNC-38; Pmyo-3::UNC-29; Pmyo-3::LEV-1; lin-15(+) (L-AChR(WT)]; ufIs58[Punc-47::RAB-3::mCherry]; oxIs19[Punc-49::UNC-49::GFP] | *Figure 7* |

## Molecular biology and plasmids

The *unc-2(zf35gf)* mutation was introduced in the *Ptag-168::UNC-2(wt)* clone (*Saheki and Bargmann, 2009*) using site-directed mutagenesis. For cell-specific *unc-2(zf35gf)* transgene expression, cell-specific promoters for GABAergic (*Punc-47*) and cholinergic (*Pacr-2*) (*Barbagallo et al., 2010*) motor neurons were amplified by PCR with FseI restriction site at the 5' end and a AscI site at the 3' end. The *Ptag-168::UNC-2(zf35gf)* construct was digested with FseI and AscI to remove the *Ptag-168* promoter and replaced with cell-specific promoters of interest. To generate the *unc-2* transgenes carrying human FHM1 mutations (*UNC-2(R192Q)* and *UNC-2(S218L))*, the UNC-2 and human CACNA1A amino acid sequences were aligned to locate the corresponding amino acid substitutions in UNC-2/CaV2. The mutations were then introduced in the *Ptag-168::UNC-2(WT)* construct by site-directed mutagenesis. The wild-type human CaV2.1 cDNA used in the HEK cell recording was obtained from Y Cao and R Tsien (*Cao and Tsien, 2010*). To generate the human CaV2.1 G1518R cDNA, UNC-2 and human CACNA1A amino acid sequences were aligned to locate the corresponding UNC-2(GF) glycine(G) to arginine (R) substitution in CACNA1A. The mutation was then introduced in the *CACNA1A* cDNA by site-directed mutagenesis.

## Isolation of *unc-2(zf35gf)* mutants, mapping and cloning

The *unc-2(zf35gf)* allele was isolated in a screen for animals that were resistant to the immobilizing effects exogenous tyramine as previously described (*Pirri et al., 2009*). We mapped *unc-2(zf35gf)* to LG X based on its hyperactive locomotion phenotype using SNP mapping (*Wicks et al., 2001*; *Davis et al., 2005*). Three-factor mapping placed *unc-2(zf35gf)* to the left of *lon-2* and close to *dpy-3*. DNA sequencing of the *unc-2* gene was performed to identify the molecular change of *unc-2(zf35gf)* mutation.

## Isolation and identification of intragenic *unc-2(zf35gf)* suppressors

*unc-2(zf35gf)* L4 animals (P0) were mutagenized with 0.5 mM N-ethyl-N-nitrosourea (ENU) for 4 hr. Approximately 10,000 F1 animals were bleached to obtain F2 eggs. F2 eggs were plated on NGM plates containing 0.25 mM aldicarb and examined for viable progeny after 7 and 14 days. Aldicarb resistant animals were individually transferred to fresh NGM plates, and their progeny were retested for aldicarb resistance. All suppressors isolated from the screen backcrossed with the wild-type N2. Suppressors that showed linkage to the X-chromosome were tested for complementation with *unc-2 (e55lf)* mutants. Molecular changes of *unc-2(zf35gf)* intragenic suppressors were identified by DNA sequencing of the *unc-2* gene.

## Behavioral and pharmacological assays

Spontaneous reversal frequency was scored on NGM plates with freshly seeded OP50. The animals were transferred from their culture plate to a new plate, and allowed to recover for 1 min. After the recovery period the number of reversals was counted for 3 min. To quantify the instantaneous velocity and average forward velocity, animals were transferred from their culture plate to a new NGM plate seeded with a thin bacterial lawn and allowed to recover for 1 min. After the recovery period, the animals were tracked for 90 s using a single worm tracker (*Yemini et al., 2013*). Videos were recorded at 30 frames per second and each frame was analyzed with worm tracking software (*Leifer et al., 2011*) to measure instantaneous velocity of single animals. Reversals, as well as 10 frames before and following each reversal, were discarded from the average forward velocity.

To examine movement defects, individual young adult worms were transferred into 96-well plates containing 50 μl M9 buffer in each well. After a 30 s recovery period, body bends were counted for 30 s. A body bend was defined as a change in direction of bending at the mid-body.

Egg-laying assays were performed as described (*Koelle and Horvitz, 1996*). Rates of egg-laying behaviors were measured by two different assays: the numbers of unlaid fertilized eggs accumulated inside of adult animals, and the developmental stages of freshly laid eggs. Briefly, in both assays, L4 larvae were isolated and allowed to develop for 40 hr. In the first method, the adults were then incubated in 96-well plates containing 1% sodium hypochlorite until the bodies were dissolved. In the second method, the adults were transferred to a fresh plate. After 30 min, the developmental stage of each freshly laid egg was determined by viewing under a high-magnification dissecting microscope.

To quantify aldicarb and levamisole resistance, young adult animals were transferred to NGM plates supplemented with 1 mM aldicarb or 0.5 mM levamisole. The percentage of paralyzed animals was scored at 15 min intervals. Animals were scored as paralyzed when they did not move when prodded with a platinum wire. To assay muscimol response, young adults were transferred onto NGM plates containing 1 mM muscimol for an hour. The rubberband phenotype was subsequently scored by analyzing the behavioral response upon touching the animal with an eyelash across its body, (posterior to the pharynx) (*de la Cruz et al., 2003*). The rubberband response was classified in 4 categories of increasing severity. : 0, animals do not contract and relax but move away from the stimulus; 1, animals contract and relax and move away from the touch stimulus; 2, animals contract and relaxed and generate a small backward displacement (less than one-half of body length); 3, animals contract and relax but fail to move backwards; 4, animals incompletely contract and relax and fail to move.

## Electrophysiology with HEK 293 cells

A stable HEK 293 cell line expressing the calcium channel auxiliary subunits β1c and α2δ (*Cao and Tsien, 2010*) was used to transiently transfect 5 μg of the wild-type or G1518R CaV2.1 α1 subunit using the calcium phosphate method. A plasmid encoding the green fluorescent protein (pGreen lantern) was also transfected to allow identification of transfected cells. Cells were cultured at 37°C in DMEM supplemented with 10% fetal bovine serum and 1000 U/ml penicillin–streptomycin.

Whole-cell inward currents were recorded 24–36 hr after transfection with a HEKA EPC-9 patch clamp amplifier. Recordings were filtered at 2 kHz and acquired using Patchmaster software (HEKA). The extracellular recording solution contained 5 mM $BaCl_2$, 1 mM $MgCl_2$, 10 mM HEPES, 40 mM TEACl, 10 mM glucose, and 87.5 mM CsCl, pH 7.4. Typically the pipettes exhibited resistances

ranging from 2 to 4 MΩ and were filled with internal solution containing: 105 mM CsCl, 25 mM TEACl, 1 mM CaCl2, 11 mM EGTA, and 10 mM HEPES, pH 7.2.

Cell capacitance (16.7 ± 6.7 pF; n = 24) and series resistance (9.7 ± 4.6 MΩ before compensation; n = 24) were measured from the current transient after a voltage pulse from −80 to −90 mV. Series resistance was typically compensated by 80–90%. Cells with large currents in which errors in voltage control might appear were discarded. I-V curves were generated by measuring the peak currents obtained after stepping the membrane potential from a holding potential of −120 mV to voltages between −55 and 40 mV in 5 mV increments for 200 ms. I-V curves were fitted with Equation 1.: $I = G(G - E_{rev})$ $(1+exp (V_{0.5}- V)/ka)^{-1}$ where G is membrane conductance, $E_{rev}$ is the reversal potential, $V_{0.5}$ is the midpoint, and $k_a$ the slope of the voltage dependence. Current densities were obtained by dividing the current peak amplitude to the cell capacitance for each experiment.

To measure steady-state inactivation profiles, conditioning pre-pulses (10 s) from −90 to 20 mV in 10 mV steps were applied, and the membrane was then stepped to the peak of the I–V curve. Currents were normalized to the maximal value obtained at the test pulse and plotted as a function of the prepulse potential. Data were fitted with Boltzmann equations: $I/I_{max}= (1 + exp[(V-V0.5)/kin]-1)$.

Data analysis was performed using the IgorPro software (WaveMetrics Inc, Lake Oswego, OR); figures, fitting and statistical analysis were done using the SigmaPlot software (version 11.0; Systat Software Inc). Data are presented as mean ± SD. Significant differences were determined using Student's t test with the significance value set at p<0.01.

## Electrophysiology with *C. elegans* neuromuscular preparations

Total spontaneous postsynaptic currents were recorded from body wall muscles as previously described (*Gao and Zhen, 2011*). Intracellular solution: K-gluconate, 115 mM; KCl, 25 mM; CaCl2, 0.1 mM; MgCl2, 5 mM; BAPTS, 1 mM; HEPES, 10 mM; Na2ATP, 5 mM; Na2GTP, 0.5 mM cAMP, 0.5 mM; cGMP, 0.5 mM. pH 7.2 with KOH,~320 mOsm. Extracellular solution: NaCl, 150 mM; KCl, 5 mM; CaCl2, 5 mM; MgCl2, 1 mM; glucose, 10 mM; sucrose, 5 mM; HEPES, 15 mM. pH 7.3 with NaOH,~330 mOsm, and the membrane potential was held at −60 mV. To isolate spontaneous excitatory postsynaptic currents, total spontaneous postsynaptic currents were recorded in *unc-49/GABA_A* receptor mutant background. To isolate spontaneous inhibitory postsynaptic currents, 0.5 mM d-tubocurarine (dTBC) was added to the extracellular solution to block acetylcholine receptors, and the membrane potential was held at −10 mV so IPSCs appeared as outward currents (*Maro et al., 2015*). All electrophysiology experiments were carried out at room temperature (20–22°C).

## Synaptic marker imaging

L4-stage transgenic animals expressing synaptic markers were picked a day before imaging. Young adults were mounted on 2% agarose pads containing 60 mM sodium azide for 5 min and immediately examined for fluorescent protein expression and localization patterns. Only animals with ventral side facing the objective were imaged. Images were captured with a Olympus BX51WI spinning disk confocal microscope with a 63x objective in the region posterior to the vulva between DD4 and DD5 neurons. Individual slices from a single animal were projected into a single image using sum projection. Defined area containing the ventral nerve cord was cropped from each image and then subjected to auto threshold for fluorescence quantification in NIH ImageJ software. Arbitrary fluorescence units of individual animal were normalized to the mean value of wild-type animals that were taken on the same day. Each n represents analysis of the nerve cord from an independent animal.

## RNAi experiments

RNAi bacteria clones were streaked on LB-Amp Tet (Amp 100 μg/ml, Tet 12.5 μg/ml) plate and grown overnight at 37°C. Single colonies were picked from these plates and grown overnight at 37°C in LB Amp 100 μg/ml. Bacteria were concentrated to 4X and seeded on NGM RNAi plates containing 6 mM IPTG and 100 μg/ml Amp. Plates were dried overnight. Six L4 worms were transferred onto desired NGM RNAi plates for 24 hr to grow to adult. Adults were transfered to another NGM RNAi plate to set up limited egg laying time intervals to obtain age-synchronized animals for analyses.

## Acknowledgements

We thank Yu-Qin Cao for CACNA1A constructs and cell lines, the *Caenorhabdits* Genetics Center (CGC), which is funded by NIH Office of Research Infrastructure Programs (P40 OD010440) for nematode strains, Andrew Leifer for Worm tracking software, Amit Sinha for RNA-seq analysis, Jan Czerminski and Micah Belew for experimental assistance and Vivian Budnik for comments on the manuscript.

## Additional information

### Funding

| Funder | Grant reference number | Author |
|---|---|---|
| National Institutes of Health | GM084491 | Mark J Alkema |
| National Institutes of Health | NS107475 | Mei Zhen |
| Canadian Institutes of Health Research | 154274 | Mei Zhen |
| Natural Sciences and Engineering Research Council of Canada | RGPIN-2017-06738 | Mei Zhen |
| National Natural Science Foundation of China | 31671052 | Shangbang Gao |
| National Institutes of Health | NS064263 | Michael M Francis |
| Consejo Nacional de Investigaciones Científicas y Técnicas | Postdoctoral fellowship | Diego Rayes |
| National Institutes of Health | NS107475 | Mark J Alkema |

The funders had no role in study design, data collection and interpretation, or the decision to submit the work for publication.

### Author contributions

Yung-Chi Huang, Conceptualization, Data curation, Investigation, Writing—original draft; Jennifer K Pirri, Shangbang Gao, Formal analysis, Investigation; Diego Rayes, Conceptualization, Formal analysis, Investigation, Writing—original draft; Ben Mulcahy, Formal analysis, Investigation, Writing—review and editing; Jeff Grant, Michael M Francis, Investigation, Writing—review and editing; Yasunori Saheki, Resources, Investigation; Mei Zhen, Supervision, Writing—original draft, Writing—review and editing; Mark J Alkema, Conceptualization, Resources, Supervision, Funding acquisition, Validation, Investigation, Writing—original draft, Writing—review and editing

### Author ORCIDs

Yung-Chi Huang https://orcid.org/0000-0001-6115-0318
Shangbang Gao http://orcid.org/0000-0001-5431-4628
Ben Mulcahy http://orcid.org/0000-0002-3336-245X
Michael M Francis http://orcid.org/0000-0002-8076-6668
Mei Zhen https://orcid.org/0000-0003-0086-9622
Mark J Alkema https://orcid.org/0000-0002-1311-5179

### Decision letter and Author response

Decision letter https://doi.org/10.7554/eLife.45905.033
Author response https://doi.org/10.7554/eLife.45905.034

## Additional files

### Supplementary files

• Transparent reporting form

DOI: https://doi.org/10.7554/eLife.45905.031

Data availability

All data generated or analyzed during this study are included in the manuscript and supporting files. Source data files have been provided for all figures.

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
