## [Decision Letter]

[Editors’ note: this article was originally rejected after discussions between the reviewers, but the authors were invited to resubmit after an appeal against the decision.]

Thank you for submitting your work entitled "Gain-of-function mutations in the UNC-2/CaV2α channel lead to excitation-dominant synaptic transmission in *C. elegans*" for consideration by *eLife*. Your article has been reviewed by a Senior Editor, a Reviewing Editor, and three reviewers. The following individuals involved in review of your submission have agreed to reveal their identity: Ronald L Calabrese (Reviewer #2).

Our decision has been reached after consultation between the reviewers. Based on these discussions and the individual reviews below, we regret to inform you that your work will not be considered further for publication in *eLife*.

As you will see from the reviews appended below, a major issue was the interpretation of the electrophysiology data. In particular, in the discussion following peer review, the concern was the conclusions based on the observed reduction in GABA synapse number without a concomitant change in mIPSC amplitude, especially since reduction in UNC-49 puncta intensity (using the same marker) has previously been shown to result in reduced mIPSC current amplitudes. Together with additional issues that were raised, the reviewers and editors concluded that this work would require extensive revisions and is better suited for a different journal.

*Reviewer #2:*

The molecular and cellular mechanisms underlying this excitatory/inhibitory (E/I) imbalance seen in disease such as familial hemiplegic migraine are poorly understood. This work identifies a gain-of-function mutation, *unc-2(gf)*, in the *Caenorhabditis elegans* CaV2 channel α1 subunit (*unc-2* gene), which leads to increased calcium currents. Such mutants exhibit locomotor and egg-laying hyperactivity. Importantly expression of the *unc-2* gene with substitutions associated with familial hemiplegic migraine lead to hyperactivity similar to that of *unc-2(gf)* mutants. *unc-2(gf)* mutants show increased cholinergic and decreased GABAergic-transmission associated with a reduction of GABAergic synapses and an increase in cholinergic synapses. The authors also present evidence that the *unc-2(gf)* mutants lead to reduction of GABA synapses in a TAX-6/calcineurin dependent manner. Thus, this work has great general interest because it provides a roadmap and an experimental system for exploring mechanistic links between Ca-channel function and changes in E/I balance in the nervous system.

The paper is clearly written, and the figures present the data needed and do so clearly.

*Reviewer #3:*

In this manuscript, Huang and colleagues, use *C. elegans* to decipher the cellular and neuronal circuit effects of a novel gain of function Ca^2+^ channel mutation. Furthermore, the authors present data to indicate that this novel mutation phenotypically resembles Ca^2+^ channel mutations found in familial hemiplegic migraine type 1. The study is significative because while it is known the in humans these types of mutations lead to increased excitation and reduced inhibition, the molecular underpinnings of this effect are unknown.

The study is in general rigorously designed; I have the following reservations though:

1) The effect of the novel mutation the authors identified on Ca^2+^ currents was studied by introducing the mutation in the human homolog. The authors do not mention anywhere the percentage of identity and homology between the worm and the human *unc-2* type channels. This is relevant because, while the loop where the mutation is located is highly conserved, other parts of the proteins may not share such a high degree of homology and they may influence the overall effect of this mutation on the function of the channel. The experiment should really be repeated by introducing the mutation in worm *unc-2*.

2) Throughout the text and in the figures, it is really hard to keep track of which gain of function mutation the authors are referring to. Is it the *unc-2(gf)* of the *unc-2/CaV2α(gf)*? Especially in light of the fact that exact parallelism between the worm and the human mutant channels function has not been firmly established, this point is particularly relevant. The authors should name the mutations throughout the text and in the figures and not simply call them *unc-2(gf)* mutants.

3) It appears that the effect of the gain of function mutation is both on voltage dependence and either trafficking or single-channel conductance (or both, at least in expression system). Indeed, currents shown in Figure 3 were recorded under conditions of complete removal of inactivation (holding -120mV) yet the mutant currents are much larger than the wild type currents. The authors need to comment on this. The authors could also compare *unc-2(wt)::GFP* and *unc-2(gf)::GFP* to see if there is more mutant protein being expressed in vivo.

4) While the results suggest that *tax-6* acts in the muscle to decrease GABAergic synapses, rescue or RNAi in muscle should confirm the conclusion of the authors.

*Reviewer #4:*

The authors analyze a novel gain of function (gf) allele of CaV2 N-type calcium channels. The *unc-2(zf35gf)* mutant was isolated in a screen for altered locomotion, lacking persistent forward or reverse movements while retaining a high level of motility. Analogous mutations in mammalian CaV2 channels expressed in tissue culture cells shows that the *zf35* mutation alters channel gating (shifting activation to more negative membrane potentials and delaying inactivation). Similar gf alleles were previously described in mammalian CaV2 and CaV1 channels. Extensive analysis of *unc-2(zf35)* mutant worms finds changes in spontaneous cholinergic and GABAergic synaptic currents, and changes in the abundance of pre-synaptic (RAB-3) and post-synaptic (UNC-49 and UNC-29) markers. Based on these studies, the authors suggest that *unc-2(zf35gf)* mutants have altered GABA synapse development and that this results from the increased excitatory transmission. These conclusions are not strongly supported by the data. I agree that transmission has changed in these mutants, but it could simply be what you expect from increased UNC-2 channel currents. The analysis of *unc-2(gf)* mutants is extensive and would likely be of interest to *C. elegans* labs; however, they do not provide a clear understanding of how nervous system development or function is altered in these mutants. For these reasons, I do not support publication in *eLife*.

Essential revisions:

1) Post-synaptic GABA receptor abundance was significantly reduced in *unc-2(gf)* mutants but this was not accompanied by a corresponding decrease in the amplitude of the spontaneous IPSCs. Several prior papers found that mutations decreasing GABA receptor puncta intensities are closely matched by similar changes in IPSC current amplitudes (e.g. Maro, Tu, and Tong papers). Given the absence of an effect on mIPSC amplitudes, these results suggest that the GABA receptors imaged here are non-synaptic. Altered non-synaptic receptors could be interesting but would require many new experiments to prove and document a functional consequence (e.g. altered tonic GABA inhibition).

2) The authors conclude that *unc-2(gf)* mutants have decreased GABA synapse numbers, based on recordings indicating a decreased mIPSC rate. The mIPSC recordings were done in the presence of a cholinergic antagonist (dTBC), which by itself decreases mIPSC rate (since cholinergic neurons provide the excitatory input to GABA neurons). It seems possible that use of dTBC biases their results toward lower mIPSC rates. I recommend recording mIPSCs without dTBC (i.e. using a low chloride internal buffer and holding potentials of 0 mV).

3) If lower synapse numbers occur in *unc-2(gf)* mutants (as the authors propose), I would expect that synaptic puncta density (both RAB-3 and UNC-49) would be reduced. The authors should report density data for the synaptic markers.

4) If the *unc-2(gf)* mIPSC rate defect cell autonomous to D neurons? If so, one would not need to propose altered cholinergic transmission in causing this phenotype (which the authors propose).

5) No controls are provided to determine if changes in synaptic puncta intensities are caused by transcriptional changes.

6) The RAB-3 puncta intensity changes are hard to interpret without further experiments. Are other SV markers altered similarly? Is the SV pool size altered (either by electrophysiology or EM)?

---

## [Author Response]

[Editors’ note: the authors’ appeal in response to the first round of peer review follows.]

Reviewer #3:In this manuscript, Huang and colleagues, use C. elegans to decipher the cellular and neuronal circuit effects of a novel gain of function Ca^2+^ channel mutation. Furthermore, the authors present data to indicate that this novel mutation phenotypically resembles Ca^2+^ channel mutations found in familial hemiplegic migraine type 1. The study is significative because while it is known the in humans these types of mutations lead to increased excitation and reduced inhibition, the molecular underpinnings of this effect are unknown.The study is in general, rigorously designed, I have the following reservations though:1) The effect of the novel mutation the authors identified on Ca^2+^ currents was studied by introducing the mutation in the human homolog. The authors do not mention anywhere the percentage of identity and homology between the worm and the human unc-2 type channels. This is relevant because, while the loop where the mutation is located is highly conserved, other parts of the proteins may not share such a high degree of homology and they may influence the overall effect of this mutation on the function of the channel. The experiment should really be repeated by introducing the mutation in worm unc-2.

We thank the reviewer for their comments. UNC-2 and mammalian CaV2 α1 subunit classes share highly similar (α1B = 73% similarity; α1A = 67%; α1E = 66%; Mathews et al. 2003). Since the similarity was published we did not include in the manuscript. We agree that this is helpful to directly compare UNC-2 and associated mutation directly with CACNA1a. Therefore, we will include an alignment of UNC-2 and CACNA1a in the supplemental figures.

The genetic data, and suppressor screen demonstrate that the *unc-2(zf35)* allele is a gain-of-function mutation of the CaV2α gene. We sought further evidence to demonstrate for the biophysical effect on channel function in cell transfection experiments. The HEK cell electrophysiology experiments show the effect of the G to R substitution on CaV2 channel function which is located in a highly conserved loop (Figure 2C). The main point of these experiments is to show that the G to R mutation results in a negative shift in voltage dependence activation, and increased current density similar to those observed for CACNA1a channels that carry FHM1 mutations. The HEK cell experiments are consistent with our behavioral-, electrophysiological and genetic analyses, clearly demonstrating that UNC-2(zf35) generates a gain-of-function phenotype. Analysis of the biophysical effect of the *zf35* mutation on the UNC-2/CaV2α channel would require the co-transfection of genes encoding the *C. elegans* ß (*ccb-1*) and α2∂ subunits (*unc-36*), and maybe other genes required for proper UNC-2 channel transport and localization *(e.g. calf-1)*. Unfortunately, no worm cell lines are available to perform these experiments. Mammalian cell lines stably transfected with ß and α2∂ subunits have been used extensively to study the effect of CaV2α mutations (including UNC-2 loss-of-function mutations; Mathews et al., 2003). This allowed us to make informative comparisons of the *zf35* mutation with those of previously characterized FHM1 mutations. The reciprocal experiment, expression of the *unc-2* transgene with corresponding human FHM1 mutations, provides strong behavioural and genetic evidence that FHM1 mutations are gain-of-function mutations.

2) Throughout the text and in the figures, it is really hard to keep track of which gain of function mutation the authors are referring to. Is it the unc-2(gf) of the unc-2/CaV2α(gf)? Especially in light of the fact that exact parallelism between the worm and the human mutant channels function has not been firmly established, this point is particularly relevant. The authors should name the mutations throughout the text and in the figures and not simply call them unc-2(gf) mutants.

The *unc-2(gf)* refers to the *unc-2(zf35)* allele. We will clarify this and refer to the gain-of-function allele as *unc-2(zf35gf)* throughout the manuscript to avoid confusion. For the protein we will refer to it as UNC-2/CaV2α(GF).

*3) It appears that the effect of the gain of function mutation is both on voltage dependence and either trafficking or single-channel conductance (or both, at least in expression system). Indeed, currents shown in Figure 3 were recorded under conditions of complete removal of inactivation (holding -120mV) yet the mutant currents are much larger than the wild type currents. The authors need to comment on this. The authors could also compare unc-2(wt)::GFP and unc-2(gf)::GFP to see if there is more mutant protein being expressed* in vivo.

We agree with the reviewer that the larger currents that we show for the CaV2α(GF) channel could reflect increased single channel conductance, enhanced expression or both. We observed an increase in current density in independent transfection experiments in HEK cells (Figure 3). We did compare expression level, and localization of the UNC-2(wt)::GFP and UNC-2(gf)::GFP in transgenic animals (Figure 2D) and did not see any obvious differences. This suggests that the mutation, besides the change in the voltage dependence of activation, also affects conductance. We will explicitly mention these possibilities in the text.

4) While the results suggest that tax-6 acts in the muscle to decrease GABAergic synapses, rescue or RNAi in muscle should confirm the conclusion of the authors.

We can perform this experiment to test the muscle requirement of *tax-6*.

Reviewer #4:The authors analyze a novel gain of function (gf) allele of CaV2 N-type calcium channels. The unc-2(zf35gf) mutant was isolated in a screen for altered locomotion, lacking persistent forward or reverse movements while retaining a high level of motility. Analogous mutations in mammalian CaV2 channels expressed in tissue culture cells shows that the zf35 mutation alters channel gating (shifting activation to more negative membrane potentials and delaying inactivation). Similar gf alleles were previously described in mammalian CaV2 and CaV1 channels. Extensive analysis of unc-2(zf35) mutant worms finds changes in spontaneous cholinergic and GABAergic synaptic currents, and changes in the abundance of pre-synaptic (RAB-3) and post-synaptic (UNC-49 and UNC-29) markers. Based on these studies, the authors suggest that unc-2(zf35gf) mutants have altered GABA synapse development and that this results from the increased excitatory transmission. These conclusions are not strongly supported by the data. I agree that transmission has changed in these mutants, but it could simply be what you expect from increased UNC-2 channel currents. The analysis of unc-2(gf) mutants is extensive and would likely be of interest to C. elegans labs; however, they do not provide a clear understanding of how nervous system development or function is altered in these mutants. For these reasons, I do not support publication in eLife.

We would like to respond to this reviewer’s comments and clarify these issues with the editors. As pointed out by the reviewer 2 and 3, the main message of the paper is that gain-of-function mutation in the presynaptic voltage gated calcium channel causes an imbalance in E/I transmission. Since the *unc-2* is expressed in all neurons, this is not what would simply be expected from increased UNC-2 calcium currents. We present evidence that increased cholinergic transmission in *unc-2/CaV2α(zf35gf)* mutants leads to a reduction of GABA synapses in a TAX-6/calcineurin dependent manner. While additional detailed molecular and cellular mechanisms remain to be explored (as with any study), our data provide a deeper understanding of how alterations in Ca-channel function produce unexpected changes in E/I balance in the nervous system. Further, our analysis of FHM mutations using this model provides evidence that these mutations produce similar changes in E/I balance that have clear implications for human disease.

Essential revisions:1) Post-synaptic GABA receptor abundance was significantly reduced in unc-2(gf) mutants but this was not accompanied by a corresponding decrease in the amplitude of the spontaneous IPSCs. Several prior papers found that mutations decreasing GABA receptor puncta intensities are closely matched by similar changes in IPSC current amplitudes (e.g. Maro, Tu, and Tong papers).

We understand the reviewer’s point as there is currently some discrepancy in the field to what extent electrophysiological data correlate with fluorescence of synaptic markers.

However, the papers cited by the reviewer do not show that decreased postsynaptic GABAR marker signals are closely matched by changes in IPSC amplitude. Instead, consistent with our findings, these studies showed that decreased GABAR intensity is closed matched by a reduction in IPSC frequency. The IPSC amplitudes, even of the same mutant, behaved differently in different preparations, and the reported changes do not strongly correlate with reported changes UNC-49::GFP fluorescence.

To summarize the relevant data from these papers:

Maro et al. (2015) showed that the degree of reduction of UNC-49 puncta signals in *madd-4, nlg-1* and *nrx-1* mutants wereclosely matched by the reduction of the IPSC frequency. For example, *nlg-1* mutants exhibited strong reduction of IPSC frequency and UNC-49 signals, but *nrx-1* did not show defects in either. Here *nlg-1* mutants exhibited a very small reduction of IPSC amplitude, which did not closely match the very strong reduction in UNC-49/GABAR clustering and fluorescence.

Tu et al. (2015) reported reduction of mIPSC frequency and amplitude in these three mutants, irrespective of their different degrees of reduction of the UNC-49/GABAR fluorescence. For example, *madd-4B(0)* showed no significant change in UNC-49 intensity, but exhibited a significant reduction of IPSC frequency, and a milder reduction in amplitude.

Tong et al. (2015) also reported that the *nlg-1* mutants exhibited reduced UNC-49 fluorescence signals, reduced mIPSC frequency, and reduced mIPSC amplitude. The reduction in mIPSC amplitude reported in Tong et al. was larger than reported in the other two studies.

Tong et al. (2015) is the only paper that reported two other mutants (*lin-2, frm-3)* that exhibited a reduction in UNC-49/GABA receptor signals, reduced mIPSC amplitude reduction but no change in mIPSC frequency. Unlike *unc-2* mutants, these two genes function strictly in muscles to regulate UNC49/GABA receptor levels. In another mutant, *nrx-1*, which was examined also in the other two papers, the authors found no overt fluorescent marker change, but an increase of mIPSC amplitude and no change in frequency. The reported mIPSC amplitude changes don’t follow an obvious trend with respect to changes in GABA receptor fluorescence intensity and are inconsistent across studies even for the same mutation.

Our results for *unc-2(gf)* mutants show concomitant changes of UNC-49/GABA receptor intensity and IPSC frequency, which is consistent with the most salient findings reported in the cited papers.

Given the absence of an effect on mIPSC amplitudes, these results suggest that the GABA receptors imaged here are non-synaptic. Altered non-synaptic receptors could be interesting but would require many new experiments to prove and document a functional consequence (e.g. altered tonic GABA inhibition).

We report a decrease in UNC-49::GFP intensity in the ventral nerve cord of *unc-2(zf35)*, a decreased frequency of mIPSCs, and an unchanged amplitude of mIPSCs. A decrease in receptor number at existing postsynaptic sites would be predicted to change amplitude whereas a loss of postsynaptic sites would be most consistent with a change in frequency. Other mutations that have been described in the literature may produce both effects, accounting for the change in amplitude and frequency. In *unc-2(gf)* mutants there appears to be a decrease in the number of postsynaptic GABAR clusters: at some NMJs, we find presynaptic of RAB-3::mCherry puncta without punctate UNC-49::GFP apposition (Figure 6—figure supplement 3). This indicates the presence of orphan presynaptic sites and post-synaptic silencing of GABA synapses. A decrease in non-synaptic receptors could account for a reduction in UNC-49::GFP intensity without changed amplitude, but does not account for the observed reduction in mIPSC frequency. A more plausible explanation, as we outline in the text, is a decrease in the number of UNC-49-containing postsynaptic sites on the muscle arms in the ventral nerve cord. This accounts for the observed reduction in mIPSC frequency, the preserved mIPSC amplitude, and the reduced UNC49::GFP intensity in the ventral nerve cord.

2) The authors conclude that unc-2(gf) mutants have decreased GABA synapse numbers, based on recordings indicating a decreased mIPSC rate. The mIPSC recordings were done in the presence of a cholinergic antagonist (dTBC), which by itself decreases mIPSC rate (since cholinergic neurons provide the excitatory input to GABA neurons). It seems possible that use of dTBC biases their results toward lower mIPSC rates. I recommend recording mIPSCs without dTBC (i.e. using a low chloride internal buffer and holding potentials of 0 mV).

We are measuring mIPSC frequency and amplitude specifically from GABAergic motor neuron (VD) synapses onto muscle. VDs are postsynaptic to cholinergic motor neurons. We include dTBC to eliminate cholinergic drive onto the VDs and isolate VD endogenous events. UNC-2(gf) is also predicted to promote cholinergic synaptic transmission onto the VDs. In the absence of dTBC, our e-phys recording would measure the compound effect of increased cholinergic drive onto VDs and VD’s endogenous IPSCs. Therefore, we believe the most clear experiment is to perform the analysis in the presence of dTBC as presented. We note that we have used this strategy to clearly distinguish mEPSCs and mIPSCs in numerous prior studies.

The recommended recording – recording mIPSCs without dTBC, with a low chloride internal buffer and holding potential of 0mV *–* has been used in several studies. We respectfully point out that this is not the only protocol used in the *C. elegans* field for recording IPSC events, and at least for us, we could not find data that demonstrated the specificity of recording conditions (data not shown in Madison, Nurrish and Kaplan, 2005).

Before applying our current electrophysiology protocol, as shown in Figure 2A-B in Maro et al. (2015), we tested the effect of holding membrane potentials and dTBCs to show that using our intracellular recording solution, at -10mV holding potential, and the in presence of 0.5mM dTBC, we blocked all EPSC events without blocking IPSC events in the muscle preparation.

For the current study, a significant increase of EPSC events (recorded in *unc-49(lf)* mutant background) in *unc2(gf)* made it more important to apply dTBC. The use of dTBC does not bias but isolate the mIPSC population from the GABAergic VD-class motor neurons in both wild-type and *unc-2(gf)* mutant for comparison.

3) If lower synapse numbers occur in unc-2(gf) mutants (as the authors propose), I would expect that synaptic puncta density (both RAB-3 and UNC-49) would be reduced. The authors should report density data for the synaptic markers.

UNC-49 abundance was quantified by calculating the overall intensity of UNC-49::GFP in the region of nerve cord imaged. This is a compound measurement, dependent on the density of clusters, intensity of each cluster, and any diffuse UNC-49::GFP that is not localized to a cluster. We also measured the cluster density, and found an increase in presynaptic RAB-3 density and decreased UNC-49 density. We can include these data in a revised manuscript. These measures parallel overall intensity measurements. We didn’t add the density measurements since they do not add much information to the overall intensity measurements. Our data are consistent with the interpretation that there are fewer GABAergic postsynapses, and orphan (silent) GABAergic presynapses.

4) If the unc-2(gf) mIPSC rate defect cell autonomous to D neurons? If so, one would not need to propose altered cholinergic transmission in causing this phenotype (which the authors propose).

We did not do this particular experiment, but we do show extensively that the effect of *unc-2(zf35)* on ventral cord UNC-49::GFP intensity is the result of a dominant effect in cholinergic neurons. Expression of *unc-2(gf)* transgene in GABAergic neurons alone, in an otherwise wild-type background, resulted in a marked increase in UNC-49::GFP intensity, whereas specific transgene expression in cholinergic neurons decreased the intensity similar to *unc-2(gf)* mutants (see Figure 7). Furthermore, cell specific expression of an *unc-2(gf)* transgene in GABAergic neurons increases aldicarb resistance, indicating an increase in GABAergic transmission. (Expression of *unc-2(gf)* in cholinergic neurons decreases aldicarb resistance). We can include these data in a revised version of the manuscript. Together, these experiments all indicate that the reduction of mIPSC rate in *unc-2(gf)* is not cell autonomous to the D-type motor neurons and further support our conclusion that altered cholinergic transmission is responsible for the transmission defects in *unc-2(gf)*.

5) No controls are provided to determine if changes in synaptic puncta intensities are caused by transcriptional changes.

We have performed transcriptional analyses but did not include these data in this manuscript. RNA-seq experiments, in which we compared wild-type vs. *unc-2(gf)* expression profiles showed no differences in *unc-49* expression but did show an increase in *unc-29* expression. We confirmed this data by qPCR. We can include these data in a revised manuscript.

6) The RAB-3 puncta intensity changes are hard to interpret without further experiments. Are other SV markers altered similarly? Is the SV pool size altered (either by electrophysiology or EM)?

This is not the main point of the paper.As noted by reviewer 2 and 3, the main point of our study is to demonstrate that gain-of-function mutations, as found in FHM1 patients, result in E/I imbalance. E/I imbalance does not simply arise due to cell autonomous excitability increases in excitatory and inhibitory neurons, but arise primarily from increased excitatory signaling onto synaptic targets, in this case muscle cells. Our electrophysiology shows that there is increased excitatory input and decreased inhibitory input to ventral muscles in *unc-2(gf)*. We provide evidence that increased cholinergic signaling leads to a reduction GABAergic synapses and explain and support this with pharmacology, fluorescent imaging of pre- and postsynaptic components of excitatory and inhibitory NMJs, cell specific expression and in vitro electrophysiology, as well as the impact on circuit output using locomotory behavior as a readout.

[Editors’ note: what follows is the author responses to the first round of peer review, after they were invited to make a revised submission.]

Reviewer #3:The study is in general, rigorously designed, I have the following reservations though:1) The effect of the novel mutation the authors identified on Ca^2+^ currents was studied by introducing the mutation in the human homolog. The authors do not mention anywhere the percentage of identity and homology between the worm and the human unc-2 type channels. This is relevant because, while the loop where the mutation is located is highly conserved, other parts of the proteins may not share such a high degree of homology and they may influence the overall effect of this mutation on the function of the channel. The experiment should really be repeated by introducing the mutation in worm unc-2.

We thank the reviewer for this comment. UNC-2 and mammalian CaV2 α1 subunit classes share a high degree of similarity. Since this was published previously (Mathews et al., 2003) we did not include in the initial version of the manuscript. We agree that this is helpful to directly compare UNC-2 and associated mutation directly with CACNA1A and now include an alignment of UNC-2 and CACNA1A in the supplemental figures (UNC-2 and CACNA1A share 68% similarity, Figure 2—figure supplement 1).

The genetic data, and suppressor screen demonstrate that the *unc-2(zf35)* allele is a gain-of-function mutation of the CaV2α gene. We sought further evidence to demonstrate the biophysical effects on channel function in cell transfection experiments. The HEK cell electrophysiology experiments show the effect of the G to R substitution on CaV2 channel function, which is located in a highly conserved loop (Figure 2C). The main point of these experiments is to show that the G to R mutation results in a negative shift in the voltage dependence of activation, and increased current density, similar to those prior observations for CACNA1A channels that carry FHM1 mutations. The HEK cell experiments are consistent with our behavioral, electrophysiological and genetic analyses, demonstrating that UNC-2(G1132R) generates a gain-of-function phenotype. Analysis of the biophysical effect of the *zf35* mutation on the UNC-2/CaV2α channel would require the co-transfection of genes encoding the *C. elegans* ß (*ccb-1*) and α2∂ subunits (*unc-36*), and maybe other genes required for proper UNC-2 channel transport and localization *(e.g. calf-1)*. Unfortunately, no worm cell lines are available to perform these experiments. Mammalian cell lines stably transfected with ß and α2∂ subunits have been used extensively to study the effect of CaV2α mutations (including UNC-2 loss-of-function mutations; Mathews et al., 2003). This allowed us to make informative comparisons of the *zf35* mutation with those of previously characterized FHM1 mutations. The reciprocal experiment, expression of the *unc-2* transgene with corresponding human FHM1 mutations, provides strong behavioral and genetic evidence that FHM1 mutations are gain-of-function mutations.2) Throughout the text and in the figures, it is really hard to keep track of which gain of function mutation the authors are referring to. Is it the unc-2(gf) of the unc-2/CaV2α(gf)? Especially in light of the fact that exact parallelism between the worm and the human mutant channels function has not been firmly established, this point is particularly relevant. The authors should name the mutations throughout the text and in the figures and not simply call them unc-2(gf) mutants.

Thank you for the suggestion. We now clarified this in the text and referred to the gain-of-function *unc* allele as *unc-2(zf35gf)* throughout the manuscript. After characterization of the *unc-2(zf35gf)* allele in the manuscript we refer to the corresponding protein as UNC-2/CaV2α(GF). UNC-2(GF) is used in figures and figure legends for brevity. The FHM1 mutations in the UNC-2 are referred to by their FHM1 amino acid substitutions (UNC-2(R192Q) OR UNC-2(S218L). We hope this avoids confusion.

*3) It appears that the effect of the gain of function mutation is both on voltage dependence and either trafficking or single-channel conductance (or both, at least in expression system). Indeed, currents shown in Figure 3 were recorded under conditions of complete removal of inactivation (holding -120mV) yet the mutant currents are much larger than the wild type currents. The authors need to comment on this. The authors could also compare unc-2(wt)::GFP and unc-2(gf)::GFP to see if there is more mutant protein being expressed* in vivo.

We agree with the reviewer that the larger currents that we show for the CACNA1A(G1518R) channel could reflect increased single channel conductance, enhanced expression or both. We observed an increase in current density in independent transfection experiments in HEK cells (Figure 3). We did compare expression level, and localization of the UNC-2(WT)::GFP and UNC-2(GF)::GFP in transgenic animals (Figure 2D) and did not see any obvious differences. This suggests that the mutation, besides the change in the voltage dependence of activation, also affects conductance. We comment on these possibilities in the Discussion section.

4) While the results suggest that tax-6 acts in the muscle to decrease GABAergic synapses, rescue or RNAi in muscle should confirm the conclusion of the authors.

We thank the reviewer for the suggestion. We have performed a *tax-6* RNAi feeding experiments. Neurons are largely resistant to RNAi feeding in *C. elegans* (Kammath et al., 2001; Timmons et al., 2001), tax-6 RNAi feeding rescued UNC-49::GFP fluorescence in *unc-2(zf35gf)* mutants. These experiments are included in the revised manuscript (Figure 7—figure supplement 3).

Reviewer #4:The authors analyze a novel gain of function (gf) allele of CaV2 N-type calcium channels. The unc-2(zf35gf) mutant was isolated in a screen for altered locomotion, lacking persistent forward or reverse movements while retaining a high level of motility. Analogous mutations in mammalian CaV2 channels expressed in tissue culture cells shows that the zf35 mutation alters channel gating (shifting activation to more negative membrane potentials and delaying inactivation). Similar gf alleles were previously described in mammalian CaV2 and CaV1 channels. Extensive analysis of unc-2(zf35) mutant worms finds changes in spontaneous cholinergic and GABAergic synaptic currents, and changes in the abundance of pre-synaptic (RAB-3) and post-synaptic (UNC-49 and UNC-29) markers. Based on these studies, the authors suggest that unc-2(zf35gf) mutants have altered GABA synapse development and that this results from the increased excitatory transmission. These conclusions are not strongly supported by the data. I agree that transmission has changed in these mutants, but it could simply be what you expect from increased UNC-2 channel currents. The analysis of unc-2(gf) mutants is extensive and would likely be of interest to C. elegans labs; however, they do not provide a clear understanding of how nervous system development or function is altered in these mutants. For these reasons, I do not support publication in eLife.

As pointed out by the reviewer 2 and 3, the main message of the paper is that gain-of-function mutation in the presynaptic voltage gated calcium channel causes an imbalance in E/I transmission. Since the *unc-2* is expressed in all neurons, this is not what would simply be expected from increased UNC-2 calcium currents. We present evidence that increased cholinergic transmission in *unc-2/CaV2α(zf35gf)* mutants leads to a reduction of GABA synapses in a TAX-6/calcineurin dependent manner. While additional detailed molecular and cellular mechanisms remain to be explored (as with any study), our data provide a deeper understanding of how alterations in Ca-channel function produce unexpected changes in E/I balance in the nervous system. Further, our analysis of FHM mutations using this model provides evidence that these mutations produce similar changes in E/I balance that have clear implications for human disease.

Essential revisions:1) Post-synaptic GABA receptor abundance was significantly reduced in unc-2(gf) mutants but this was not accompanied by a corresponding decrease in the amplitude of the spontaneous IPSCs. Several prior papers found that mutations decreasing GABA receptor puncta intensities are closely matched by similar changes in IPSC current amplitudes (e.g. Maro, Tu, and Tong papers).

We understand the reviewer’s point as there is currently some discrepancy in the field to what extent electrophysiological data correlate with fluorescence intensity of synaptic markers. We do believe that the reviewer’s statement does not accurately represent the results from the cited papers.

Specifically, the papers cited by the reviewer did not show that decreased postsynaptic GABAR marker signals are closely matched by changes in IPSC amplitude. Instead, and consistent with our findings, these studies showed that decreased GABA receptor marker intensity is closed matched by a reduction in IPSC frequency. The IPSC amplitudes, even of the same mutant, behaved differently in different preparations, and reported changes do not strongly correlate with changes in UNC-49::GFP fluorescence.

To summarize the relevant data from these papers:

Maro et al. (2015) showed that the degree of reduction of UNC-49 puncta signals in *madd-4, nlg-1* and *nrx-1* mutants wereclosely matched by the reduction of the IPSC frequency. For example, *nlg-1* mutants exhibited strong reduction of IPSC frequency and UNC-49 signals, but *nrx-1* did not show defects in either. Here *nlg-1* mutants exhibited a very small reduction of IPSC amplitude, which did not closely match the very strong reduction in UNC-49/GABAR clustering and fluorescence.

Tu et al. (2015) reported reduction of mIPSC frequency and amplitude in these three mutants, irrespective of their different degrees of reduction of the UNC-49/GABAR fluorescence. For example, *madd-4B(0)* showed no significant change in UNC-49 intensity, but exhibited a significant reduction of IPSC frequency, and a milder reduction in amplitude.

Tong et al. (2015) reported that the *nlg-1* mutants exhibited reduced UNC-49 fluorescence signals, reduced mIPSC frequency, and reduced mIPSC amplitude. The reduction in mIPSC amplitude reported in was larger than reported in the other two studies. Tong et al., (2015) reported two other mutants (*lin-2, frm-3)* that exhibited a reduction in UNC-49/GABA receptor signals, reduced mIPSC amplitude reduction but no change in mIPSC frequency. Unlike *unc-2*, these two genes function strictly in muscles to regulate UNC-49/GABA receptor levels. In another mutant, *nrx-1*, which was also examined in the other two papers, the authors found no overt fluorescent marker change, but an increase of mIPSC amplitude and no change in frequency. The reported mIPSC amplitude changes don’t follow an obvious trend with respect to changes in GABA receptor fluorescence intensity and are inconsistent across studies, even for the same mutation.

Our results for *unc-2(zf35gf)* mutants show concomitant changes of UNC-49/GABA receptor intensity and IPSC frequency, which is consistent with the most salient findings reported in the cited papers.

Given the absence of an effect on mIPSC amplitudes, these results suggest that the GABA receptors imaged here are non-synaptic. Altered non-synaptic receptors could be interesting but would require many new experiments to prove and document a functional consequence (e.g. altered tonic GABA inhibition).

We report a decrease in UNC-49::GFP intensity in the ventral nerve cord of *unc-2(zf35)*, a decreased frequency of mIPSCs, and an unchanged amplitude of mIPSCs. A decrease in receptor number at existing postsynaptic sites would be predicted to change amplitude whereas a loss of postsynaptic sites would be most consistent with a change in frequency. Other mutations that have been described in the literature may produce both effects, accounting for the change in amplitude and frequency. In *unc2(zf35gf)* mutants there appears to be a decrease in the number of postsynaptic GABAR clusters: at some NMJs, we find presynaptic of RAB-3::mCherry puncta without punctate UNC-49::GFP apposition (Figure 6—figure supplement 3). This indicates the presence of orphan presynaptic sites and post-synaptic silencing of GABA synapses. A decrease in non-synaptic receptors could account for a reduction in UNC-49::GFP intensity without changed amplitude, but does not account for the observed reduction in mIPSC frequency. As we outline in the text, we believe that a decrease in the number of UNC-49containing postsynaptic sites on the muscle arms in the ventral nerve cord is a more plausible explanation. This accounts for the observed reduction in mIPSC frequency, the preserved mIPSC amplitude, and the reduced UNC-49::GFP intensity in the ventral nerve cord.

2) The authors conclude that unc-2(gf) mutants have decreased GABA synapse numbers, based on recordings indicating a decreased mIPSC rate. The mIPSC recordings were done in the presence of a cholinergic antagonist (dTBC), which by itself decreases mIPSC rate (since cholinergic neurons provide the excitatory input to GABA neurons). It seems possible that use of dTBC biases their results toward lower mIPSC rates. I recommend recording mIPSCs without dTBC (i.e. using a low chloride internal buffer and holding potentials of 0 mV).

We are measuring mIPSC frequency and amplitude specifically from GABAergic motor neuron (VD) synapses onto muscle. VDs are postsynaptic to cholinergic motor neurons. We include dTBC to eliminate cholinergic drive onto the VDs and isolate VD endogenous events. The UNC-2(GF) channel is also predicted to promote cholinergic synaptic transmission onto the VDs. In the absence of dTBC, our electrophysiology recording would measure the compound effect of increased cholinergic drive onto VDs and VD’s endogenous IPSCs. Therefore, we believe the clearest experiment is to perform the analysis in the presence of dTBC as presented. We note that we have used this strategy to clearly distinguish mEPSCs and mIPSCs in prior published studies.

The recommended recording – recording mIPSCs without dTBC, with a low chloride internal buffer and holding potential of 0mV *–* has been used in several studies. But we respectfully point out that this is not the only protocol used in the *C. elegans* field for recording IPSC events. At least for us, we could not find published data that demonstrated the specificity of stated recording conditions. The closest we found was “data not shown” in Madison et al., 2005.

Before applying our electrophysiology protocol (Maro et al., 2015; Figure 2A-B), we tested the effect of holding membrane potentials and dTBCs to show that using our intracellular recording solution, at -10mV holding potential, and the in presence of 0.5mM dTBC, we blocked all EPSC events without blocking IPSC events in the muscle preparation.

For the current study, a significant increase of EPSC events (recorded in *unc-49(lf)* mutant background) in *unc-2(zf35gf)* made it more important to apply dTBC. The use of dTBC does not bias but isolate the mIPSC population from the GABAergic VD-class motor neurons in both wild-type and *unc-2(gf)* mutant for comparison.

3) If lower synapse numbers occur in unc-2(gf) mutants (as the authors propose), I would expect that synaptic puncta density (both RAB-3 and UNC-49) would be reduced. The authors should report density data for the synaptic markers.

UNC-49 abundance was quantified by calculating the overall intensity of UNC-49::GFP in the region of nerve cord imaged. This is a compound measurement, dependent on the density of clusters, intensity of each cluster, and any diffuse UNC-49::GFP that is not localized to a cluster. We also measured the cluster density, and found an increase in presynaptic RAB-3 density and decreased UNC-49 density. We now include these data in the revised manuscript (Figure 6—figure supplement 2). These measures parallel overall intensity measurements. Our data are consistent with the interpretation that there are fewer GABAergic postsynapses, and orphan (silent) GABAergic presynapses.

4) If the unc-2(gf) mIPSC rate defect cell autonomous to D neurons? If so, one would not need to propose altered cholinergic transmission in causing this phenotype (which the authors propose).

We did not do this particular experiment, but we do show extensively that the effect of *unc-2(zf35gf)* on ventral cord UNC-49::GFP intensity is the result of a dominant effect in cholinergic neurons. Expression of *unc-2(zf35gf)* transgene in GABAergic neurons alone, in an otherwise wild-type background, resulted in a marked increase in UNC-49::GFP intensity, whereas specific transgene expression in cholinergic neurons decreased the intensity similar to *unc-2(zf35gf)* mutants (see Figure 7). Furthermore, cell specific expression of an *unc-2(zf35gf)* transgene in GABAergic neurons increases aldicarb resistance, indicating an increase in GABAergic transmission. (Expression of *unc-2(zf35gf)* in cholinergic neurons decreases aldicarb resistance). We now include these data in a revised version of the manuscript (Figure 7—figure supplement 2). Together, these experiments all indicate that the reduction of mIPSC rate in *unc-2(zf35gf)* is not cell autonomous to the D-type motor neurons and further support our conclusion that altered cholinergic transmission is responsible for the transmission defects in *unc-2(zf35gf)*.

5) No controls are provided to determine if changes in synaptic puncta intensities are caused by transcriptional changes.

We thank the reviewer for pointing this out. We did perform transcriptional analyses but did not include these data in this manuscript. RNA-seq experiments, in which we compared expression profiles of wild-type vs. *unc-2(zf35gf)* animals showed no significant differences in *unc-29, unc-49, myo-3, rab-3, unc-2* and *rgef-1* expression. We include these data in a revised manuscript (Figure 6—figure supplement 4).

6) The RAB-3 puncta intensity changes are hard to interpret without further experiments. Are other SV markers altered similarly? Is the SV pool size altered (either by electrophysiology or EM)?

We agree that future studies are needed to provide additional mechanistic insights. However, this is not the main point of the paper.As noted by reviewer 2 and 3, the main point of our study is to demonstrate that gain-of-function mutations, as found in FHM1 patients, result in E/I imbalance. E/I imbalance does not simply arise due to cell autonomous excitability increases in excitatory and inhibitory neurons, but arise primarily from increased excitatory signalling onto synaptic targets, in this case muscle cells. Our electrophysiology shows that there is increased excitatory input and decreased inhibitory input to ventral muscles in *unc-2(zf35gf)*. We provide evidence that increased cholinergic signalling leads to a reduction GABAergic synapses and explain and support this with pharmacology, fluorescent imaging of pre- and postsynaptic components of excitatory and inhibitory NMJs, cell specific expression and in vitro electrophysiology, as well as the impact on circuit output using locomotory behavior as a readout.